# Simulations of the Holocene Climate in Europe Using an Interactive Downscaling within the iLOVECLIM model (version 1.1)

*Frank Arthur[1], Didier M. Roche[2,3], Ralph Fyfe[4], Aurélien Quiquet[3], Hans Renssen[1]*

[1]Department of Natural Sciences and Environmental Health, University of South-Eastern Norway, Bø, Norway.

[2]Faculty of Science, Cluster Earth and Climate, Vrije Universiteit Amsterdam, Amsterdam, the Netherlands.

[3]Laboratoire des Sciences du Climat et de l'Environnement, LSCE/IPSL, CEA-CNRS-UVSQ, Université Paris-Saclay Gif-sur-Yvette, France.

[4]School of Geography, Earth and Environmental Sciences, University of Plymouth, Plymouth, UK.

*Correspondence to*: Frank Arthur (Frank.Arthur@usn.no)

**Abstract.** This study presents the application of an interactive downscaling in Europe using iLOVECLIM (a model of intermediate complexity), increasing its atmospheric resolution from 5.56° to 0.25° kilometric. A transient simulation using the appropriate climate forcings for the entire Holocene (11.5 – 0 kyr BP) was done for both the standard version of the model and with an interactive downscaling applied. Our results show that simulations from downscaling present spatial variability which agrees better with proxy-based reconstructions and other climate models as compared to the standard model. The downscaling scheme simulates much higher (by at least a factor of two) precipitation maxima and provides detailed information in mountainous regions. We focus on examples from the Scandes Mountains, the Alps, the Scottish Highlands and the Mediterranean. The higher spatial resolution of the downscaling provides a more realistic overview of the topography, gives local climate information such as precipitation and temperature gradient that is important for paleoclimate studies. With downscaling, we simulate similar trends and spatial patterns of the precipitation changes reconstructed by other proxy studies (for example in the Alps), as compared to the standard version. Our downscaling tool is numerically cheap, implying that our model can perform kilometric-multi-millennial simulations and is suitable for future studies.

# 1 Introduction

Numerical climate models are used to study the past, present and future climate change, two types of global climate models are primarily used, the so-called General Circulation Models (GCMs) and Earth System Model of Intermediate Complexity (EMICs). GCMs and EMICs simulate the climate of the Earth by applying mathematical equations to describe the atmospheric, oceanic and land interactions or feedbacks. These climate models are evaluated with past climate data to ensure that their sensitivity to climate change is realistic, and thus improve their ability to project future climate change. GCMs and EMICs have been used to simulate the past climate. Examples are the Last Glacial Maximum (e.g., Liu et al., 2021), the Holocene (e.g., Claussen et al., 2002; Renssen et al., 2009; Renssen and Osborn, 2003; Schmidt et al., 2004; Otto-Bliesner et al., 2006), and the Last Millennium (e.g., Crowley, 2000; Jones et al., 2001; Zorita et al., 2005). These paleoclimate simulations have been routinely compared with proxy-based paleo data to evaluate their performance (Masson et al., 1999; Bonfils et al., 2004; Brewer et al., 2007; Bartlein et al., 2011). However, the difference in spatial resolution between the simulated climate model results and proxy-based paleo reconstructions make this comparison problematic and usually poses some uncertainties (Renssen et al., 2001; Ludwig et al., 2019). Transient simulations of the Holocene with GCMs are still currently a challenge due to the numerical cost (it can take more than 4-5 months). Therefore, EMICs (like iLOVECLIM) have simplified physics and are computationally more efficient, making it feasible to perform large ensemble experiments at a multi-millennial timescale, which is an advantage to paleoclimate studies. EMICs can simulate explicitly the interaction between all the components of an earth system model and simulate the transient and equilibrium climate sensitivity (Claussen et al., 2002). Still, EMICs' representation of the large-scale atmospheric moisture content and other processes produced by local scale features such as mountains ranges, water bodies, forest etc. is quite poor and thus affecting the dynamics of these sub-components that rely on the global atmospheric water cycle (Quiquet et al., 2018). This implies that many of the processes that govern the local climate (vegetation, hydrology and topography) are not well represented in most EMICs' coarse resolution.

This limitation of global climate models can be overcome by applying spatial downscaling, a primary tool in meteorology and climate studies. Downscaling can establish a relationship between large-scale atmospheric processes and the local scale to derive information at a fine spatial resolution (Castro et al., 2005). There are two approaches of downscaling used to resolve this coarse-fine resolution variance: statistical and dynamical downscaling (Murphy, 1999). Statistical downscaling (SDM) involves creating an empirical relationship between historical large-scale atmospheric characteristics (such as pressure fields) and local climate variables (temperature, precipitation, etc.), and applying these statistical relationships to the output of large-scale global variables (GCMs/EMICs) to simulate the local climate variables (e.g., Stoner et al., 2013). The main types of SDMs are "weather typing" methods, which are based on conditioning the simulations of small-scale data on recurrent weather types over a specific region (e.g. Willems and Vrac, 2011), "transfer functions", which link large-scale atmospheric conditions and local-scale data directly (e.g. Vrac et al., 2007), and "stochastic weather generators", that generates local-scale time series from their possibly conditional probability density functions (e.g. Olsson et al., 2009). There are some studies related to statistical downscaling in a paleo perspective. For example, Latombe et al. (2018) simulated the climate of the Last Glacial Maximum (23-19 thousand years before present, kyr BP) over western Europe by statistically downscaling the temperature and precipitation time series outputs from a GCM with a generalized additive model (GAM). Lorenz et al. (2016) performed a transient paleoclimate simulation with statistical downscaling for North America spanning between the period 21 kyr BP to 2100 AD at 0.5° spatial resolution. Their study provides datasets which offer a standard collection of climate simulations that may be used to model the impact of past and future climate change on biodiversity.

Dynamical downscaling is the technique used by global models to simulate the land-atmosphere interaction process by considering the sub-grid, orography, and other conditions over a local scale (Feser et al., 2011). Dynamical downscaling is thus aimed at increasing the spatial resolution (horizontally and vertically) by simulating the regional sub-component of the

climate from global models (Ludwig et al., 2019). In contrast, statistical downscaling only assumes constant statistical relationships between large and local-scale processes. These relationships may not be valid for conditions very different from the present, such as in the early Holocene. Conversely, since it is based on physical laws, dynamical downscaling can be applied to any period and gives more comprehensive information for some specific regions, particularly if precipitation is highly influenced by local topography (Gomez-Navarro et al., 2011; Wang et al., 2015; Raible et al., 2017; Quiquet et al., 2018). However, there may be some uncertainties and limitations in the use of dynamical spatial downscaling. The uncertainties of downscaled temperature and precipitation usually depend on the errors associated with the large-scale model such as its physical parameterization (Murphy, 1999; Feser et al., 2011), the model's simplification or limitation and the biases associated with the model's atmospheric circulation (Quiquet et al., 2018).

Comparing paleoclimate model results at a high spatial resolution with proxy-based data informs meaningful interpretation (Bonfils et al., 2004; Russo and Cubasch, 2016; Ludwig et al., 2019). Dynamical downscaling has been applied to both present and future climate analyses to give an improved estimate of future climate change (e.g., Jacob et al., 2007). According to Jacob et al. (2014), regional (dynamical) downscaling better represents the physical processes that trigger precipitation and provides more realistic outputs in complex regions when compared to outputs from low-resolution models. Despite these advantages, applying dynamical downscaling in paleoclimate studies is still limited, with the main reason being that it is computationally intensive and costly to run long-term climate simulations with downscaling. Still, there is good potential, as comparing paleoclimate simulations with proxy-based reconstructions is more meaningful at a high spatial resolution.

Some attempts have been made in the past to simulate high-resolution climate in a paleo-perspective. Examples are available for the LGM (Yokoyama et al., 2000; Strandberg et al., 2011; Hofer et al., 2012; Ludwig et al., 2016) and the Little Ice Age/Medieval Warm Period over arid central Asia (Fallah et al., 2016). The Last Millennium over the Iberian Peninsula was studied with a Regional Climate Model (RCM) of 30 km spatial resolution to evaluate the significance of the internal variability of temperature and precipitation at regional scales (Gomez-Navarro et al., 2011). Renssen et al. (2001) also applied an RCM to simulate the European climate during the Younger Dryas cold period (12.9-11.7, kyr BP). In addition, Gomez-Navarro et al. (2013) performed a higher resolution RCM simulation in Europe spanning from 1500 – 1990, their results compared to observed CRU datasets show an improvement in the climate model most particularly in areas of complex topography. In a different study, the same results were compared with empirical proxy-based reconstructions (Gomez-Navarro et al., 2015), and the results show regional mean biases, particularly for summer temperature and winter precipitation. These biases were similar to biases found relative to CRU observational data in their earlier study (e.g., Gomez-Navarro et al., 2013). Russo and Cubasch (2019) present a dynamical downscaling for different time slices of mid – to – late Holocene in Europe using the RCM model COSMO – CLM. Comparison of their results with observed CRU data show that, their RCM can reproduce a realistic climatology. The same model was recently applied by Russo et al. (2022) to analyze summer temperatures of the mid-Holocene in Europe and to understand the potential causes of the discrepancy between results of climate model simulations and pollen-based reconstruction (Russo et al., 2022). Moreover, Velasquez (2021) applied dynamical downscaling to the LGM climate of Europe and utilized the land and atmospheric components of GCM (CCSM4), RCM (WRF) and dynamic vegetation model simulations to study the significance of land–atmospheric response for the European climate. The results from the study suggest that, the regional climate is significantly influenced by the LGM land cover (Velasquez, 2021).

The Holocene (hereafter 11.5-0 kyr BP) is a significant period for studying the climate evolution and variability to improve our knowledge of the climate system. The period is well known and archived with proxy data (e.g., Masson et al., 1999; Bonfils et al., 2004; Braconnot et al., 2007; Wanner et al., 2008; Mauri et al 2015). It is also a period used to evaluate how climate models respond to the variations in insolation in response to astronomical forcing (Fischer and Jungclaus, 2011). In the early

Holocene, the astronomical forcing was very different from today because of changes in three astronomical parameters (eccentricity, obliquity, and the precession) which alter the amount and distribution of incoming solar energy at the top of the atmosphere (Berger, 1978). According to Berger (1978), during the early Holocene at 11 kyr BP, the Northern Hemisphere
received about 30 W/m$^2$ more insolation during boreal summer than at present, which caused the climate to be relatively warm during the early to mid-Holocene. This relatively high summer insolation resulted in the reorganization of various variables of the climate system, such as the melting of the ice sheets (including the Fennoscandia Ice Sheet, FIS and Laurentide Ice Sheet, LIS), and the associated freshwater release in different regions during the early Holocene (Briner et al., 2016; Zhang et al., 2016).


Within the Holocene, the mid-Holocene (MH, 6 kyr BP) is a time-slice that was characterized by relatively warm conditions in the Northern Hemisphere, associated with the astronomically forced summer insolation. Other forcings were similar to pre-industrial values, including atmospheric greenhouse gas concentration levels (Bartlein et al., 2011). The MH is very well documented, offering the opportunity to study this warm condition and long growing season in some regions such as Europe.
Pollen-based climatic variables such as surface temperature over Europe (Mauri et al., 2015) show that the climate system also responded to other large scale complex processes such as the orography or land-surface interactions with the atmosphere, and atmospheric circulation (Bonfils et al., 2004). Previous climate model studies, when compared to proxy data for the mid Holocene, suggest that some climate models simulate a cooler and wetter climate in Southern Europe during winter but with a very weak signal (e.g., Brewer et al., 2007; Brayshaw et al., 2011). However, there is a poor agreement for Southern Europe
between proxy-based reconstructions of summer temperature and almost all climate models. Climate models show warming in summer in the Mediterranean region at 6 kyr BP relative to the pre-industrial era (e.g., Masson et al., 1999; Mauri et al., 2014; Fischer and Jungclaus, 2011; Russo and Cubasch, 2016; Brierley et al., 2020). This model response is tightly connected to the insolation change at mid-Holocene, with summer insolation being higher and winter insolation being lower. The modelled MH temperatures reflect this, including over the Mediterranean region, while continental-scale reconstructions
from pollen records show large cooling during summer over the Mediterranean region (Davis et al., 2003; Mauri et al., 2015).

Several modelling groups have recently performed transient modelling simulations of the Holocene. For instance, Bader et al., (2020) used the Max Planck Institute for Meteorology Earth System Model (MPI-ESM 2) to run two different global Holocene transient simulations for temperature which span from 6000 BCE to 1850 CE. Their results conclude with one simulation showing global cooling during the late Holocene while the other simulation shows global warming in the late Holocene. The
results also indicate that the warming is most conspicuous in the tropics. Paleoclimate proxies indicate that the Northern Hemisphere summer temperature was warmer during the first part of the Holocene than the pre-Industrial era, primarily because the perihelion occurred during the boreal summer. However, most global-scale proxy-based reconstructions are inconsistent with long-term warming simulated by climate models in response to ice sheet retreat and rising greenhouse gas concentration throughout the Holocene epoch, a discrepancy termed the 'Holocene temperature conundrum' (Liu et al., 2014).
Whiles reconstructions show a cooling trend, model results suggest a warming trend. Accordingly, one of the most striking differences between climate reconstructions and climate model simulations is the direction of global temperature change in the Holocene. Furthermore, the accuracy of LGM and Holocene paleoclimate simulations depends on detailed knowledge of paleoclimate boundary conditions, which may not be independent from proxies.

Within this study, we present a simulation of the transient Holocene climate evolution during the last 11.5-0 kyr BP in Europe (Fig 1), performed with both the standard version of the iLOVECLIM model (Roche et al., 2014) and a version with an interactive downscaling (Quiquet et al., 2018). The downscaling is performed at each model time step, and it is consistent between the two grids, with the precipitation at the coarse resolution interacting with the sub-grid. The spatial resolution in

Europe is increased from 5.6° latitude × 5.6° longitude to 0.25° latitude × 0.25° longitude. Our general objective is to examine the impact of the downscaling on the model results and evaluate if the model (with downscaling) simulates the climate during the Holocene in better agreement with other climate models and proxy data. Thus, our main goal is to evaluate the benefits of using downscaling in paleo-climate simulation. In addition, we will assess the impact of the model resolution on their sensitivity at high spatial resolution. We wish to provide a comprehensive and consistent overview of the climate system at a fine resolution during the Holocene.

We will answer the following questions in this paper:

(i)     What is the impact of downscaling on the precipitation patterns during the Holocene in different, Regions in Europe?

(ii)    Do the high-resolution results of precipitation in the mountainous regions (e.g., the Alps, the Scandes and the Scottish Highlands) produce Holocene climate patterns that compare more favorably to proxy data and other climate models when compared with the low-resolution results?

(iii)   What is the advantage of using cheap numerically interactive downscaling for paleoclimate research?

## 2 Model, Simulations and Methods

### 2.1 The ILOVECLIM Model

iLOVECLIM (hereafter version 1.0) (Roche et al., 2014) is a three-dimensional model, with a simplified representation of the atmosphere relative to GCMs. This simplification and its lower spatial resolution make iLOVECLIM much faster than coupled GCMs (Goosse et al., 2010; Kitover et al., 2015). It is a fork of the LOVECLIM 1.2 model code (Goosse et al., 2010), and it has the main climate system components in common. We apply here a version that includes the following components: the atmospheric model, ECBilt (Opsteegh et al., 1998), the sea-ice ocean component, CLIO (Goosse and Fichefet, 1999), and the terrestrial vegetation model, VECODE (Brovkin et al., 1997).

Our model version is a direct follow-up of the ECBilt-CLIO-VECODE coupled climate model and has been successfully used to simulate some key past and future climates, for example the LGM (e.g., Timmermann et al., 2004; Roche et al., 2007), the last deglaciation (e.g., Timm et al., 2008), the Holocene (e.g., Renssen et al., 2005a, b), the last millennium (e.g., Goosse et al., 2005a, b) and the 21st Century (e.g., Schaeffer et al., 2002, 2004; Driesschaert et al., 2007). The atmospheric component (ECBilt) includes three vertical levels at 800, 500 and 200 hPa and applies the quasi-geostrophic potential vorticity equation to model the dynamical processes in the atmosphere (Opsteegh et al., 1998). It runs on a global spectral grid truncated at T21 that represents a horizontal resolution of 5.6° latitude and 5.6° longitude. Another component of the model, CLIO, Coupled Large-scale Ice-Ocean model (Goosse et al., 2010), is a three-dimensional free-surface Ocean General circulation model which has been coupled with a full sea ice model (Goosse and Fichefet, 1999). It has a horizontal resolution of 3° by 3° lat-lon and 20 layers in the vertical. VECODE runs on the same grid as ECBilt and includes three different plant functional types (PFTs): trees, grass, and desert/bare soil, each with different physical properties for evapotranspiration, surface roughness and albedo. The vegetation fraction ($v$) is calculated by the sum of the tree fraction ($f$) and grass fraction ($g$) (Goosse et al., 2010).

### 2.2 Interactive Downscaling

In this study, we apply an interactive downscaling presented by Quiquet et al. (2018) for precipitation and temperature in the coupled iLOVECLIM model. The downscaling is done from the original ECBilt's T21 grid towards a European domain between 13.875° W and 49.875° E in longitude and 35.125° N and 71.875° N in latitude with a resolution of 0.25° in lat-lon.

The basic idea behind the downscaling process is to reproduce the model physics of ECBilt (not the dynamics) on a higher spatial resolution so that the sub-grid orography is explicitly considered. To do so, we use artificial vertical layers, so that variables such as temperature and precipitation formation can be computed at any altitude in the sub-grid orography for each atmospheric time step (Quiquet et al., 2018). We follow a conservative approach in which the "large scale" fields (on the native grid) are the sum/mean of what is computed on the sub-grid. First, the downscaling is performed at each model time step during run time. Secondly, there is a two-way coupling between the coarse grid and the sub-grid which ensures consistency (the precipitation at the coarse resolution is the sum of the sub-grid precipitation). As such, there is a strong difference with standard offline downscaling techniques. The results from Quiquet et al. (2018) show that, in comparison to the standard version of the model, the downscaling improves the vertical distribution of temperature (for example, with a more realistic profile in mountainous regions) and the precipitation distribution in mountainous regions. However, the results also suggest that the downscaling is not able to correct the biases of the large-scale native model, which are mostly driven by the model's simplification and to the atmospheric circulation, which is not downscaled (Quiquet et al., 2018).

To apply interactive downscaling in the model, the temperature and moisture variables on the vertically extended native (ECBilt) grid are recomputed in the model. The computation is done on the 11 vertical levels of the grid (10, 250, 500, 750, 1000, 1250, 1500, 2000, 3000, 4000 and 5000 m) (Quiquet et al., 2018), by using the equations required for the vertically extended grid defined by Haarsma et al. (1997). The atmospheric boundary layer is not well represented in the ECBilt, hence the heat and moisture fluxes at the earth surface are computed based on an idealized vertical profile (Quiquet et al., 2018). The temperatures are computed based on hydrostatic equilibrium and the ideal gas law at the 650 and 350-hPa horizon, with the assumption that the atmosphere is isothermal above 200 hPa. Above 500 hPa, the atmosphere is assumed dry (Goose et al., 2010). The temperatures and precipitation at the sub-grid orography are then computed from the climate variables obtained or computed from the vertically extended artificial grids (Quiquet et al., 2018). For detailed information, such as an explanation of the physics applied on the downscaling in the model, the reader is referred to Quiquet et al. (2018).

## 2.3 Experimental set-up

We applied iLOVECLIM-1.0 (Roche et al., 2014) and iLOVECLIM-1.1 (Quiquet et al., 2018) to simulate the transient evolution of the climate during the last 11.5 kyr (Table 1). Two experiments were performed (hereafter 11.5K_Standard and 11.5K_Down). The experiment 11.5K_Standard is performed with the standard version of the model on the low-resolution T21 grid. The second experiment (11.5K_Down) is when downscaling has been applied to the quasi-geostrophic T21 grid to compute the temperature and precipitation on the regional sub-grid in Europe (Fig. 1). We forced the simulations with orbital forcings (Berger, 1978) and atmospheric trace gas concentrations of $CO_2$, $CH_4$ and $N_2O$ (Raynaud et al., 2000) which vary annually for the 11.5-0 kyr BP simulation period. Constant ice sheet configurations were prescribed for the experiments. While the solar constant and aerosol levels were kept fixed at pre-industrial levels. During the Holocene, the astronomical forcing determines variations in terms of seasons and latitudes of the incoming solar radiation at the top of the atmosphere. For example, at 65°N, the summer insolation is reduced by 30 W/m$^2$ throughout the Holocene epoch (Fig. 2). The ice core-based levels of $CO_2$, $CH_4$ and $N_2O$ represent 1 W/m$^2$ variability in radiative forcing (Schilt, 2010). This greenhouse gas (GHG) forcing was at its maximum level at 10 kyr BP and then started decreasing to a low value at 8 kyr BP before rising again during the last 6 kyr BP to preindustrial values (Fig. 2). The experiments were initialized with a state derived from an experiment that was run for 1000 years until equilibrium with 11.5 kyr BP astronomical parameters, greenhouse gas levels, and ice sheets.

We present the results of precipitation and temperatures as anomalies relative to the (pre-industrial period at the end of our simulations) and compare our down-scaled results (11.5K_Down) with the results of the standard version (11.5K_Standard).

# 3 Results

## 3.1 Spatial distribution of annual temperature anomalies and annual precipitation anomalies in Europe

Simulated annual temperature anomalies relative to the pre-industrial at 11.5K_Down and 11.5K_Standard (Fig. 3) show some
similarities in terms of their spatial patterns. However, as expected more details are visible in 11.5K_Down than in
11.5K_Standard. The downscaling produces local temperature changes, visible on the high-resolution grid (11.5K_Down)
particularly in northern Scandinavia and the Alps at both 9 kyr BP and 6 kyr BP (Fig. 3 d & e). Simulated annual temperature
anomalies for 11.5K_Down and 11.5K_Standard were warm at both 9 kyr BP and 6 kyr BP with annual temperature anomalies
(relative to pre-industrial) of up to 4 °C for 9 kyr BP and 2 °C for 6 kyr BP. Central Europe and South-West Europe have
positive temperature anomalies relative to the preindustrial, reaching between 0.5-1°C at 9 kyr BP. Only south Turkey has
negative temperature anomaly up to -2 °C at 9 kyr BP. During the mid-Holocene, northern Scandinavia was 2 °C warmer than
the pre-industrial, whereas northern Europe was relatively warm with an annual temperature anomaly of 0.5 °C. The south-
eastern corner of the domain was cool with a negative annual temperature anomaly of -1 °C at 6 kyr BP. At 3 kyr BP, most
regions in Europe were cooler than the pre-industrial except for the south-western part, which had a positive surface
temperature anomaly of up to 0.5 °C. The latitudinal pattern during the mid-Holocene shows that the warming was stronger
at the high latitudes than the mid latitudes.

These spatial patterns in our results during the mid-Holocene appears to agree with PMIP4/CMIP6 related work analyzed by
Williams et al. (2020) who found in Europe mean annual temperature anomalies between 1 to 2 °C warmer than the pre-
industrial, which is similar to our results. Our model simulates cooler conditions in south-eastern Europe but slightly higher
temperatures in the south-west compared to the pre-industrial, which contradicts the cool conditions in the south-west
suggested by Brewer et al. (2007) based on reconstructions of proxy data. Reconstructions of mean annual temperature in the
mid-Holocene by Wu et al. (2007) reveal a similar pattern in some regions to our results, showing intense cooling in southern
Europe and warming over northern and central Europe. However, our model simulates similar magnitude of warming relative
to the pre-industrial over northern Scandinavia, which is more in agreement with Mauri et al. (2015).

Overall, the native grid (T21/11.5_Standard) is still seen in the 11.5K_Down model results in many regions for all times slices.
This is because the main impact of the dynamical downscaling is to compute physically the climate variables which are related
to temperature in accordance with the sub grid topography for a given course-grid information.

The simulated mean precipitation anomalies show that 11.5K_Down provides more spatial detail than 11.5K_Standard, and
better considers the impact of topography on precipitation (Fig. 4). The results reveal that 11.5K_Down drastically increases
the spatial variability of the model in topographic regions when compared to 11.5K_Standard. For instance, our 11.5K_Down
experiment provides a more detailed view in the Scandes Mountains, the Alps, the Scottish Highlands and the Pyrenees. The
Scandes mountains are characterized by wetter than pre-industrial conditions at 9 and 6 kyr BP for the 11.5K_Down, but
relatively drier conditions at 9, 6 and 3 kyr BP than the pre-industrial for 11.5K_Standard (Fig. 4). The annual precipitation
anomaly in the Scandes Mountains at 9 kyr BP was about 250 mm/yr for 11.5K_Down, while the 11.5K_Standard has a mean
annual precipitation anomaly of -50 mm/yr. The annual mean precipitation anomaly relative to the pre-industrial in the mid
Holocene (6 kyr BP) was up to 50 mm/yr and -50 mm/yr in Scandinavia for 11.5K_Down and 11.5K_Standard respectively.
In the Alps, the Pyrenees and the Massif Central, the 11.5K_Down simulated annual precipitation anomalies at 9 kyr BP were
up to 150 mm/yr higher than the pre-industrial, and this detailed information is not seen in 11.5K_Standard (Fig. 4). Even at 6
kyr BP, our 11.5K_Down precipitation provides additional information in the Alps with an annual precipitation anomaly up
to 50 mm/yr. The Scottish Highlands show precipitation of about 350 mm/yr up to 400 mm/yr wetter than the pre-industrial in

the early Holocene (9 kyr BP) for 11.5K_Down. However, it was about 50 mm/yr drier than the pre-industrial in the late Holocene (3 kyr BP) for both the 11.5K_Down and 11.5K_Standard experiments in the Scottish Highlands. The Scottish Highlands were still up to 100 mm/yr wetter in the mid-Holocene than the pre-industrial for 11.5K_Down; this was approximately 50% less in 11.5K_Standard that shows precipitation anomalies between 50 mm/yr up to 100 mm/yr. The mountain ranges thus drastically affect the local precipitation anomalies, eventually changing the sign of the standard model (e.g., the downscaled Alps and the Scandes are most of the time in opposition with the standard model. Wetter regions becomes drier and drier regions becomes wetter). The higher precipitation anomalies in these mountainous regions are due to the impact of the downscaling, as the primary effect of the downscaling is to increase precipitation in elevated areas (e.g., the Scandes Mountains and the Alps). The results with downscaling provide details of the precipitation that better reflect the effect of the underlying topography. In general, experiment 11.5K_Down is relatively wetter than 11.5K_Standard in most topographically complex regions in Europe,

The simulated annual mean precipitation anomalies with respect to the pre-industrial for the 11.5K_Down grid reproduce some of the major large-scale structures in Europe. For instance, the annual precipitation anomalies of 11.5K_Down show a pattern characterized by a relatively dry zone in central Europe, which splits wetter areas south and east of the Mediterranean (Morocco, Algeria, Turkey and Middle East) from wetter north-west Europe, especially at 9 kyr BP and 6 kyr BP (Fig. 4). At 9 kyr BP, south-west Iberia and southern Turkey were wetter with precipitation anomalies ranging from 100 mm/yr to 400 mm/yr and showing more spatial details. This is generally true for 11.5K_Standard but the spatial details do not provide much information compared to the 11.5K_Down grid. North-west Europe have precipitation anomalies of about 50 mm/yr to 300 mm/yr at 9 kyr BP. In contrast, the dry zone in central Europe is characterized by a negative precipitation anomaly of up to -50 mm/yr at 9 kyr BP. The impact of the downscaling is seen in the results, as the downscaling reproduces some local topographical features in these regions. The results in 11.5K_Down suggest that northern Italy was about 50 mm/yr drier relative to the pre-industrial. However, in 11.5K_Standard, northern Italy was relatively up to 50 mm/yr wetter than the pre-industrial. Other regions such as north-east Europe (especially western Russia) were generally dry (about -50 mm/yr) throughout the Holocene for the 11.5K_Down grid at 9 and 6 kyr BP but were relatively wet (up to 50 mm/yr) for 11.5K_Standard. The Iberian Peninsula in the 11.5K_Down grid was between 50 mm/yr and 100 mm/yr wetter than pre-industrial at 9 Kyr BP, but up to -50 mm/yr drier than pre-industrial for the simulated 11.5K_Standard.

## 3.2 Temporal Trends in Annual Precipitation for the Holocene in Europe

In most areas, applying an interactive downscaling leads to an increase in precipitation compared to the standard experiment. The average precipitation in Europe for the entire Holocene was 775 mm/yr and 624 mm/yr for 11.5K_down and 11.5K_Standard respectively. Consequently, this shows about 24% increase in precipitation for the whole of Europe when downscaling is applied. In both experiments, it was generally wetter in the early and mid-Holocene than the pre-industrial especially between 10 and 7 kyr BP. For the 11.5K_Down model, precipitation generally rises from 762 mm/yr at 11 kyr BP to its maximum peak value of 822 mm/yr between 9 and 8.5 kyr BP, and slightly decreases after 7.5 kyr BP to 746 mm/yr in the late Holocene (Fig. 5). This precipitation trend in the Holocene is similar to 11.5K_Standard. Precipitation was about 622 mm/yr at 11 kyr BP for 11.5K_Standard, then rises steadily to 666 mm/yr between 9 and 8.5 kyr BP before declining gradually to 593 mm/yr towards the pre-industrial. The mid-Holocene was 32 mm/yr wetter than the pre-industrial in the 11.5K_down. The precipitation has a decreasing trend towards the pre-industrial.

**3.3 Regional Annual Precipitation evolution for the Holocene (Scandes Mountains, Alps, and Scottish Highlands)**

One of our objectives in this study is to evaluate our model's performance for regions with elevated topography. We compare the precipitation from reconstructed proxy data over the Alps, the Scandes Mountains, and the Scottish Highlands with our model results.

**Scandes Mountains**

Compared to 11.5K_Standard, our 11.5K_Down produces about twice as much precipitation in the Scandes Mountains, with an opposite long-term trend. In the downscaled version, precipitation rises gradually from 1233 mm/yr at 11 kyr BP to its maximum peak of 1469 mm/yr around 9 kyr BP (Fig. 6), after which precipitation started declining gradually to 1 kyr BP. This contrasts with 11.5K_Standard, which has rising precipitation trend from 525 mm/yr at 11 kyr BP to the pre-industrial level of 637 mm/yr (Fig. 6 a). In addition. there is a clear maximum peak at 9 kyr BP in 11.5K_Down that is absent in 11.5K_Standard.

**Alps**

Similar to the Scandes Mountains, there is a doubling of the precipitation in 11.5K_Down when compared to 11.5K_Standard in the Alps. The trends for both experiments are similar with reduced precipitation in the early Holocene, and a flat trend afterwards (Fig. 6). The precipitation trend in the Alps for 11.5K_Down shows that these mountains were drier in the early Holocene when compared to pre-industrial, with the late Holocene showing a flat trend towards the pre-industrial. Thus, there is higher precipitation in the late Holocene than the early Holocene for 11.5K_Down. The figure for 11.5K_Down shows a slight increase of precipitation in the early Holocene towards 7 kyr BP followed by a slight dip and stable trend towards pre-industrial. For 11.5K_Standard, the precipitation rises from 11 kyr BP was quite steady and stable to the late Holocene with less variability when compared to 11.5K_Down.

**Scottish Highlands**

In the Scottish Highlands, 11.5K_Down model simulates the highest average precipitation in Europe with a Holocene mean value of 3238 mm/yr, compared to 1077 mm/yr for 11.5K_Standard. Thus, the downscaled result is about three times higher than the 11.5K_Standard. Generally, precipitation rises from 11.5 Kyr BP to its maximum peak average of 3474 mm/yr around 8.7 Kyr BP and declines gradually through the rest of the Holocene for 11.5K_Down (Fig. 6 f). Compared to pre-industrial, 11.5K_Down simulates wetter conditions in the early and mid-Holocene, but after 3.6 kyr BP the model simulates a stable trend. The precipitation pattern for 11.5K_Standard is quite different from 11.5K_Down. The 11.5K_Standard experiment shows gradual decline towards pre-industrial while the decrease in 11.5K_Down is more pronounced with its maximum peak at a different time.

**4 Discussion**

The precipitation values simulated by 11.5K_Down clearly show the influence of topography, since the highest values of above 400 mm/yr are produced in mountainous regions such as the Scandes, Scotland and the Alps, which have a much higher elevation. The basic effect of the interactive downscaling is to redistribute precipitation in a physically consistent way based on topography (Quiquet et al., 2018). In most cases, 11.5K_Standard is wetter than 11.5K_Down in less elevated regions (Fig. 4). For example, some parts of central Europe are relatively dry in 11.5K_Down at 9 kyr BP but are relatively wet in 11.5K_Standard. Since the topography in 11.5K_Down is more realistic, the spatial pattern obtained and the distribution with the downscaling is better than in the standard version. Thus, downscaling reproduces local features of these mountain regions

described in the results, with higher precipitation in agreement with what is known from modern observations. The annual and selected regional precipitation trends presented in the results reveal that all the selected regions in the 11.5K_Standard experiment present less precipitation. In comparison, 11.5K_Down simulates much higher precipitation, coinciding with topography variations.

Our 11.5K_Down simulation represents climate at a regional scale closer to the spatial scale of proxy data than our 11.5K_Standard experiment. The improvement resulting from the downscaling technique will impact the comparison of our results with other climate model simulations and proxy-based reconstructions, especially because the latter are influenced by local conditions that are more realistically represented in the model. Previous model-data comparisons have revealed that General Circulation Models (GCMs) have great difficulty simulating key Holocene climate features, particularly trends in southern Europe (Mauri et al., 2014). One important factor could be the coarse resolution of GCMs (about 200-600 km) relative to the regional and local climate represented by proxy records. Thus, these models may not be able to account for a fine-scale variability of local features such as complex topography. To evaluate the performance of our model, we have compared our 11.5K_Down and 11.5K_Standard annual precipitation results for some regions in Europe with available proxy data and other simulated climate models. Some studies suggest that proxy-based reconstructions could be biased towards the growing season (Bader et al., 2020). However, studies using inverse vegetation modelling have found no evidence for seasonal biases in pollen-based reconstructions (Davis et al. 2017; Chevalier et al., 2020). For example, Davies et al. (2017) shows comparisons with modern analogue technique (MAT) and inverse modelling (IVM) of proxy-based reconstructions for the mid-Holocene in the Mediterranean and their findings show no significant difference in the methods, which implies no evidence of bias in the MAT. Given these results, we did not consider any impact of a seasonal bias.

The trend for precipitation reconstructions of Mauri et al. (2015) in the Alps shows an increase in precipitation from early to mid-Holocene, similar to our 11.5K_Down experiment but do not show such a significant increasing trend (Fig. 7a). In terms of the precipitation temporal trend, our 11.5K_Down agrees with lake-level reconstructions by Harrison et al. (1996) that suggest no clear Holocene trend in lake-level records derived from some high-altitude sites (above 1000 m) in the Alps such as Landos and Rousses. Although our model simulations do not match perfectly with the proxy-based reconstructions of Mauri et al. (2015), the downscaled trend with reduced precipitation in the early Holocene is in better agreement with the proxy-based precipitation trend than our standard version. The same is true for the Scandes Mountains (Fig. 7b) where the downscaled results show a maximum in precipitation between 10 and 8 kyr BP followed by a decreasing trend, in agreement with the proxy-based reconstructions (Mauri et al., 2015). To provide an example of the advantages of the applied downscaling, we assess the performance of the simulations at a high-elevated site in the Italian Alps (Armentarga peat bog, 2345 m asl) with proxy-data reconstructions based on pollen data (Furlanetto et al., 2018). The 11.5K_Down and the reconstructions show a middle to late Holocene precipitation increase (Fig. 9b). The level of Holocene precipitation reconstructed by Furlanetto et al. (2018) for the high elevated region is between 1100 and 1600 mm/yr, which is less but closely agrees with our 11.5K_Down simulation that simulates precipitation between 1500 and 1750 mm/yr (Fig. 9b). Their reconstructions thus suggest that precipitation in this elevated topographic region was higher than the 650 mm/yr simulated by our 11.5K_Standard (Fig. 9a), which supports the higher values that can be seen in the 11.5K_Down simulations. It is therefore likely that the 11.5K_Standard result is less realistic than 11.5K_Down in oceanic mountain regions. The 11.5K_Standard result of 650 mm/yr is more appropriate for precipitation of the surrounding lowlands, which would be expected as the standard experiment does not take into consideration the topography in this site in the Alps.

Comparing our 11.5K_Down with the 11.5K_Standard version of the simulations for the Scandes Mountains, 11.5K_Down shows that the early Holocene (10 - 5 kyr BP) was a wetter period which is followed by drier conditions in the late Holocene,

while the 11.5K_Standard result is characterized by flat precipitation pattern with no significant trend (Fig. 6). Similar to the Alps, proxy data support our results with downscaling, as proxy-based precipitation reconstructions from Scandinavia suggest a more humid and wet early Holocene, followed by a dry mid to late Holocene (Seppä and Birks, 2001; Bjune et al., 2004; Harrison et al., 1996). For example, the pollen-based climate reconstructions by Seppä and Birks (2001) have in Scandinavia a similar trend as our 11.5K_Down results, showing enhanced precipitation in the early Holocene, which decreased steadily

towards the late Holocene. This pattern is not seen in the 11.5K_Standard simulations. Bjune et al, (2005) reconstructed winter precipitation based on Holocene glacier behavior and show drier conditions in Scandinavia from 11.5 to 8 kyr BP, a wetter period from 8 to 4 kyr BP, followed by a drier period after 4 kyr BP to pre-industrial. The pattern of their results thus agrees with the 11.5K_Down simulation (Fig. 6). Moreover, pollen-inferred precipitation anomalies from Mauri et al. (2015) show that the spatial pattern in their precipitation in the Scandes Mountains is similar to our downscaling experiment presented in

Fig. 4. Consequently, our 11.5K_Down results is in better agreement with the proxy-based reconstructions than our 11.5K_Standard in most available studies and shows that downscaling also provides a more realistic representation of the hydroclimate in Scandinavia. However, it should be noted that our model simulations assumed present-day topography and did not correct precipitation for post-glacial isostatic uplift in the Scandes mountains, potentially leading to an overestimation of precipitation in the Scandes earlier in the Holocene.


The comparison with proxy-based reconstructions in our third region, the Scottish Highlands, is hampered by the unavailability of suitable records. However, a comparison to modern data makes clear that the downscaling results are much more representative of the precipitation in the high-altitude Scottish Highlands than the standard results. Our precipitation values in this region for the high resolution was overestimated by the model, but we can have some confidence that 11.5K_Down

represents the Holocene precipitation conditions also better than 11.5K_Standard. due to the spatial variability.

We also compared our 11.5K_Down results with studies in the Mediterranean. This region is very sensitive to changes in humidity, and during the early and mid-Holocene period the dominant controlling factor on the Mediterranean ecosystem was precipitation rather than temperature (Magny et al., 2013; Mauri et al., 2015; Peyron et al., 2017). For instance, Brayshaw et

al. (2011) used the HadSM3 global climate model, which was dynamically downscaled to about 50 km using a regional climate model (HadRM3), to simulate enhanced precipitation for time slices during the Holocene in the Mediterranean. This is consistent with the results of our 11.5K_Down high-resolution model, which simulates wetter conditions during most of the Holocene relative to pre-industrial (Fig 4). Their regional climate model simulations show that some coastal areas particularly in the north-eastern Mediterranean received more precipitation at 9 kyr BP and 6 kyr BP (Brayshaw et al., 2011). This agrees

with the high-resolution model that simulates higher precipitation above 450 mm/yr around the Balkans and Southern Turkey. The 11.5K_Down results also show a contrasting pattern between different regions in the Mediterranean, with the southern and eastern Mediterranean being wet and the western-central part being dry (Fig. 4), particularly during the early-to-mid Holocene. This pattern is similar to reconstructions based on proxy data, such as lake levels, pollen data, and stable isotopes. All these data show that throughout the Holocene, climate conditions in the Mediterranean region varied spatially and

temporally (e.g., Mauri et al., 2015; Sadori et al., 2016; Cheddadi and Khater, 2016). For instance, an east–west division during the Holocene is also observed in the Mediterranean region from lake-level reconstructions (Magny et al., 2013), marine and terrestrial pollen records (Guiot and Kaniewski, 2015) and speleothem isotopes (Roberts et al., 2011). Specifically, the simulated wetter mid-Holocene conditions in 11.5K_Down agree with the high precipitation reconstructed at 6 kyr BP (Bartlein et al. 2011; Guiot and Kaniewski, 2015; Mauri et al., 2015; Kuhnt et al., 2008). In the Mid-Holocene, precipitation

values of 100-500 mm/yr higher than the pre-industrial levels were reconstructed in the Mediterranean by Bartlein et al. (2011), in agreement with the high-resolution simulation (Fig. 4). Pollen-based reconstructions by Peyron et al. (2017) suggest dry early to mid-Holocene conditions in northern Italy similar to the downscaling simulations. The synthetic multi-proxy

reconstructions of Finne et al. (2019) for the Holocene in the Mediterranean show a longer period of wetter conditions in the east and south as compared to the north and central areas of the Mediterranean. This was especially clear before 8.7 kyr BP. Their study also reveals that the driest period in the eastern Mediterranean was at 3 kyr BP, while Italy remained wetter around this time (Finne et al., 2019). Comparing their work with our 11.5K_Down, we can see some similarities. For instance, at 3 kyr BP, the eastern Mediterranean was the driest compared to the early Holocene and mid-Holocene, and these drier conditions agree with the reconstructions from their study.

As shown above, we can compare our high-resolution results with paleo-reconstructions in these complex mountainous terrains in Europe due to the spatial variability attained from the downscaling. Compared to other studies, we find that the downscaled precipitation simulations are consistently in line with some European regions. Europe experienced multiple climate changes of various magnitudes over the Holocene, and regions within the continent experienced those changes differently. Based on our results, we can reproduce the different regional responses presented by some proxy-based reconstructions, for instance in northern and southern Europe, we find wetter conditions from the early to mid-Holocene relative to pre-industrial (Fig. 4), similar to the proxy-based reconstructions reported by Mauri et al. (2015).

We have compared our results with some of the climate models from the Paleoclimate Modeling Intercomparison Project 4 (PMIP4). For example, Williams et al. (2020) simulated a mid-Holocene climate in Europe that was wetter than the pre-industrial with precipitation anomalies between 200-400 mm/yr. The magnitude of the simulated anomalies of their study is similar to the proxy reconstructions of Bartlein et al. (2011) with proxy-based anomalies between 200-400 mm/yr. Comparing the 11.5K_Down simulations to Williams et al. (2020), we find that our results agree with some regions, but are in contrast with some other regions. For example, at 6 kyr BP, our model simulates wetter conditions relative to the pre-industrial with anomalies above 200 mm/yr in the eastern Mediterranean. However, our model simulates drier conditions in the eastern part of Europe towards Russia, which is in contradiction with their studies. The mid-Holocene mean annual precipitation results of the PMIP4 ensembles in the mid-Holocene studied by Brierley et al. (2020) show positive precipitation anomalies in the Mediterranean similar to our 11.5K_Down results. However, the PMIP4 ensemble mean does not capture the slight increase of precipitation in northern and central Europe and much wetter conditions in the Mediterranean as reconstructed for the mid-Holocene by Bartlein et al. (2011). Our simulations suggest slightly drier conditions in north-eastern Europe in agreement with the PMIP4 ensembles. Previous studies suggest that the persistent mismatch between model simulations and reconstructions is usually due to the biases in the pre-industrial control (Harrison et al., 2015). As studied by Brierley et al. (2020) in the PMIP4 (15 ensemble simulations), different climate models would usually give different results, but this may have no impact on the general conclusions.

One important variable for describing past climate is the intensity of the growing season as expressed by growing-degree days above 0°C (GDD0). Our model simulates more optimal conditions for plant growth at 6 kyr BP relative to pre-industrial in some parts of central Europe and northern Europe, most particularly in Scandinavia (Fig. 8) between 150 – 200 degree days. The growing degree days decrease in southern and eastern Europe, with a vast decrease in Turkey, Russia, Spain and Italy. The model simulations fit well with the proxy-based reconstructions of Mauri et al. (2015), which show an increase in growing season intensity in Scandinavia and a decrease in southern Europe, particularly in Turkey (shown in the supplementary of Mauri et al., 2015). Jiang et al. (2018) analysed the growing season over ice free land for the mid-Holocene based on numerical simulations from 28 PMIP 2 and 3 models. Their results show a latitude-varying difference in the mid-Holocene relative to the pre-industrial. For example, 24 out of their 28 analysed PMIP model results indicate that growing season prolonged from the northern mid-to high latitude (above 50°N) and shorted between (20° – 50°N), similar to our results. It is well documented that the most important forcing for the mid-Holocene is the orbitally induced incoming solar radiation at the top of the

atmosphere in the Northern Hemisphere (Berger, 1978), which is stronger in summer than at the pre-industrial. The longer growing seasons in the North are a response to this orbital forcing. This was also found by Bartlein et al. (2011), who reconstructed the mid-Holocene changes in growing degree days using subfossil-pollen and plant-macrofossil data. Their results show an increase in growing degree warmth over northern Europe and a decrease over southern Europe most particularly around the Mediterranean. The study by Bartlein et al. (2011) was supported with previous reconstructions based on different approaches for instance, through inverse modelling (Wu et al., 2007), a modern analogue method based on lake level data (Cheddadi et al., 1996) and climate calibration from different plant functional types (Tarasov et al., 1999). Our model simulations agree with these proxy-based reconstructions in most regions in Europe.

The high-resolution in our simulations shows spatial details for precipitation in the Holocene. In terms of climate impact studies, such local scale information achieved from downscaling can be very useful. However, with our high-resolution simulations, only the physical part of precipitation is downscaled. The main reason for precipitation increases over the mountains with the downscaling is simply because it is colder at high elevation and cold air cannot contain much humidity, so it rains out. But since we see more of the topography, it is better represented in the model (Quiquet et al., 2018). We argue that such high-resolution climate modeling can be very useful to paleoclimatologists for model-data comparison at the local scale. For example, if a scientist studying paleoclimate data retrieves data from high-elevated region such as the Alps or the Scandes Mountains, then it would be highly useful to get information on the gradients in precipitation and temperature provided by a high-resolution (regional) climate model. It is probable that these regional proxy data have been impacted by these gradients, hence it can be expected that the spatial scale of the high-resolution (regional) climate model will be in better agreement with the proxy data than the coarse (global) resolution model.

The interactive downscaling method that we have used for this study is relatively simplified since only the atmospheric physics are downscaled according to the sub-grid orography. The atmospheric dynamics remains computed at the coarse grid scale. As such, the fine-scale structure of the wind pattern that could affect the precipitation and temperature is not accounted for. For example, high precipitation on windward slopes and equally low precipitation on leeward slopes cannot be reproduced. One way to improve the model in future development would be to downscale the atmospheric dynamics in addition to the atmospheric physics. However, this will require heavy developments given the spectral nature of the atmospheric grid. An alternative solution would be to weigh the local sub-grid precipitation by the native grid wind direction. To do so, we could use some pre-computed index based on the normal vector of the sub-grid orography surface. Also, as discussed earlier, the downscaling tends to redistribute the precipitation according to the elevation with an increase in precipitation at high elevation and a respective drying at low elevation. As a result, although the pattern of precipitation appears better reproduced, there is an overestimation of the precipitation in most high elevation regions (e.g., Scottish Highlands). This bias is a direct consequence of the physics of precipitation in iLOVECLIM where the precipitation is mostly the results of the local humidity. In the model, when this humidity reaches a critical fraction of the water vapour saturation, rain or snow is produced. Since the saturation is non-linearly linked to temperature, for a given humidity the precipitation rate is much higher at high elevation where the temperature is colder. On top of this aspect, the original model biases can also largely explain the biases of the downscaled version. For example, iLOVECLIM has a large warm bias at the global scale which favours a humid atmosphere.

**5 Conclusions**

In this study, we have applied an interactive downscaling to our low-resolution iLOVECLIM model in Europe, increasing its resolution from 5.56° to 0.25° latitude-longitude. To our knowledge, this is the first-time downscaling has been applied for Holocene transient experiments. A transient simulation for the entire Holocene (11.5 – 0 kyr BP) was done for both the standard version of the model and with downscaling being applied. We have answered the following research questions in this paper:

*What is the impact of dynamical downscaling on the precipitation patterns during the Holocene in different complex regions in Europe?* We have compared the spatial and temporal annual precipitation results of the low-resolution grid with the high-resolution grid to analyze the impact of downscaling on the model. Our results suggest that when downscaling is applied for precipitation, it drastically increases the spatial variability particularly in high-elevated regions as compared to the coarse resolution of the standard model.

*Are the high-resolution results of precipitation in the mountainous regions (e.g., the Alps, the Scandes Mountains and the Scottish Highlands) producing more realistic Holocene climate when compared with the low-resolution grid and other proxy data?* We have shown that the high-resolution simulation presents a better agreement with proxy-based reconstructions and other climate model studies as compared to the course (low) resolution grid particularly in the Mediterranean and mountainous regions in Europe. The downscaling scheme simulates much higher (by at least a factor of two) precipitation maxima and provides detailed information in the Scandes Mountains and the Alps. By comparing our 11.5K_Down and 11.5K_Standard simulations with published proxy-based reconstructions, 11.5K_Down simulates in close agreement the magnitude of the precipitation changes reconstructed by other proxy studies (for example high elevated sites), and that there is good agreement for the overall trend and spatial pattern than 11.5K_Standard. The different patterns of change such as wetter conditions in northern and southern Europe are well captured by our 11.5K_Down model. Overall, precipitation was higher during the early Holocene than the late Holocene in most regions in Europe when compared to the pre-industrial.

*What is the advantage of using a numerically cheap interactive downscaling in paleoclimate research?* Paleoclimatologists would like to have very high-resolution model runs covering the last million years or more. We have shown a numerically cheap tool which is able to perform multi-millennial simulations at a km scale, and even with a low resolution and a simple downscaling scheme, we achieve relatively good model-data agreement. The downscaling technique is moderately computationally demanding, making it appropriate for long-term integration. It can hypothetically be applied and extended in further studies to any resolution higher than the T21 grid. Our downscaling produces more detailed precipitation information suitable for comparison with regional paleoclimate studies. In addition, the downscaling simulations are better suited to match proxy data in terms of spatial representation, making the downscaling a useful approach for comparisons between climate models and proxy data. The downscaling's improved ability to resolve complex topography areas is very important since proxy records are often obtained from high altitudes, where the most sensitive climate archives (such as trees, sediments and ice cores) are found.

**Acknowledgement**

We thank Marie José Gaillard who provided input in setting up this work. We also thank Heikki Seppä and Basil Davis, who assisted us with proxy-based reconstructions. Finally, we want to give special thanks to the editor (Laurie Menviel) and the two anonymous reviewers for their comments and suggestions, which helped to improve the paper.

**Code Availability**

The iLOVECLIM source code is accessible at http://www.elic.ucl.ac.be/modx/elic/index.php?id=289 (UCL, 2021). The developments on the iLOVECLIM source code are hosted at http: //forge.ipsl.jussieu.fr/ludus (IPSL, 2021), due to copyright restrictions they cannot be publicly accessed. Request for access can be made by contacting D. M. Roche (didier.roche@lsce.ipsl.fr). For this study, we used the model at revision 1147.

**Data Availability**

https://doi.org/10.23642/usn.19354082

**Author contribution**

All authors designed the study. FA performed the simulations and wrote the manuscript with contributions of HR, RF, DMR and AQ. The model results were analysed and interpreted by all authors.

**Competing interests:** None.

**Financial Support**

The research is financed through the European Union's Horizon 2020 research and innovation programme within the TERRANOVA project, No 813904. The paper reflects the views only of the authors, and the European Union cannot be held responsible for any use which may be made of the information contained therein.

**Review statement.** This paper was edited by Laurie Menviel and reviewed by two anonymous referees.

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

**Tables and Figures**

Table 1: Summary of the main features of the model and experimental set-up

| simulations | | |
|---|---|---|
| **Model** | **ILOVECLIM (Standard Version)** | **iLOVECLIM (downscaling)** |
| **Component** | Ocean, sea ice, atmosphere, vegetation | |
| **Atmospheric Resolution (lat × lon)** | 5.6° × 5.6° | 0.25° × 0.25° |
| **Oceanic component Resolution** | 3° × 3° | |
| **Prescribed forcings and reference** | Orbital forcings     Berger (1978) <br><br> GHG     Schilt et al., (2010) <br>        Raynaud et al., (2000) <br> Icesheets, Fixed | |
| **Initial condition** | Equilibrium experiment at 11.5 ka (1kyr) | |
| **Duration of experiment** | 11.5 kyr | |


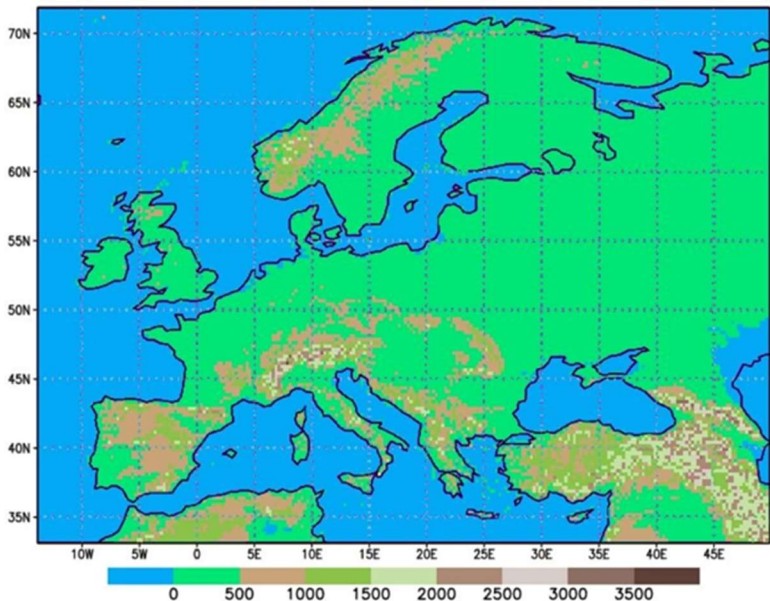

Figure 1. The topography of Europe as included in the interactive downscaling.

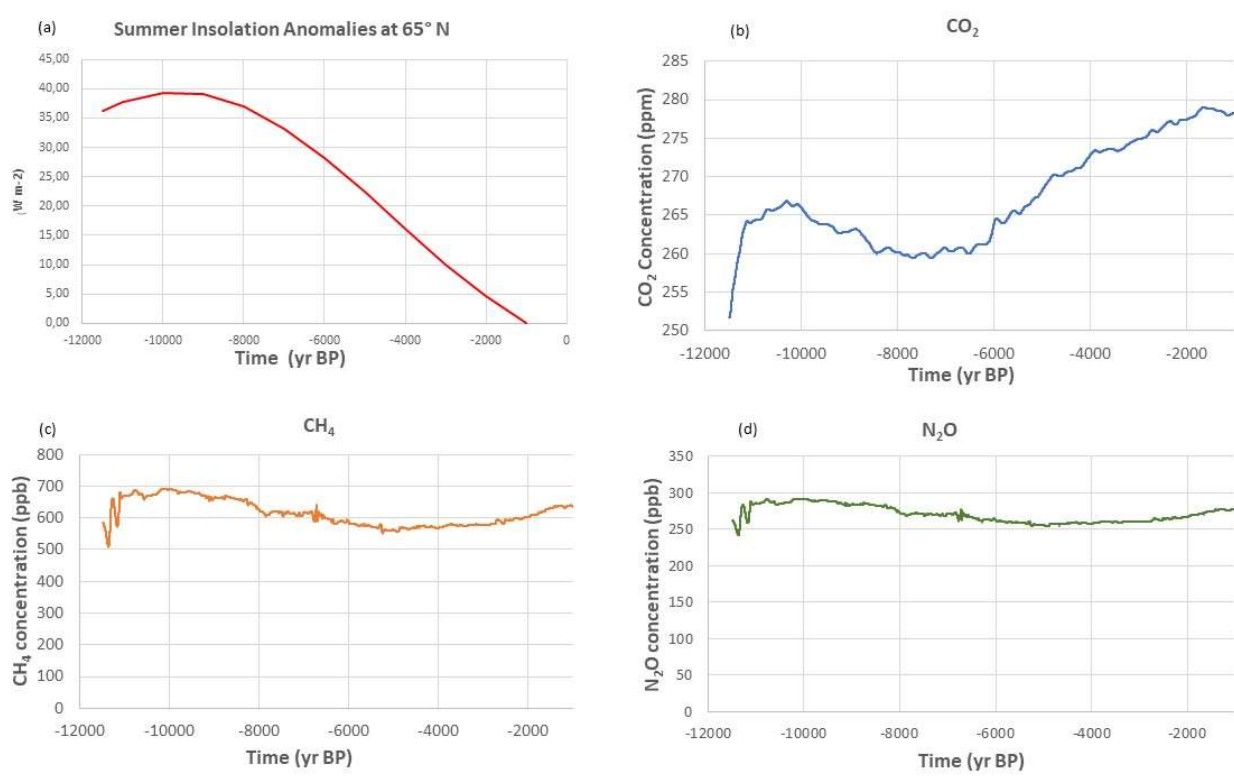


Figure 2. Climate forcings used in the experiment GHG forcings (Raynaud et al., 2000) and summer (July) insolation at 65° N during the Holocene (Berger, 1978).

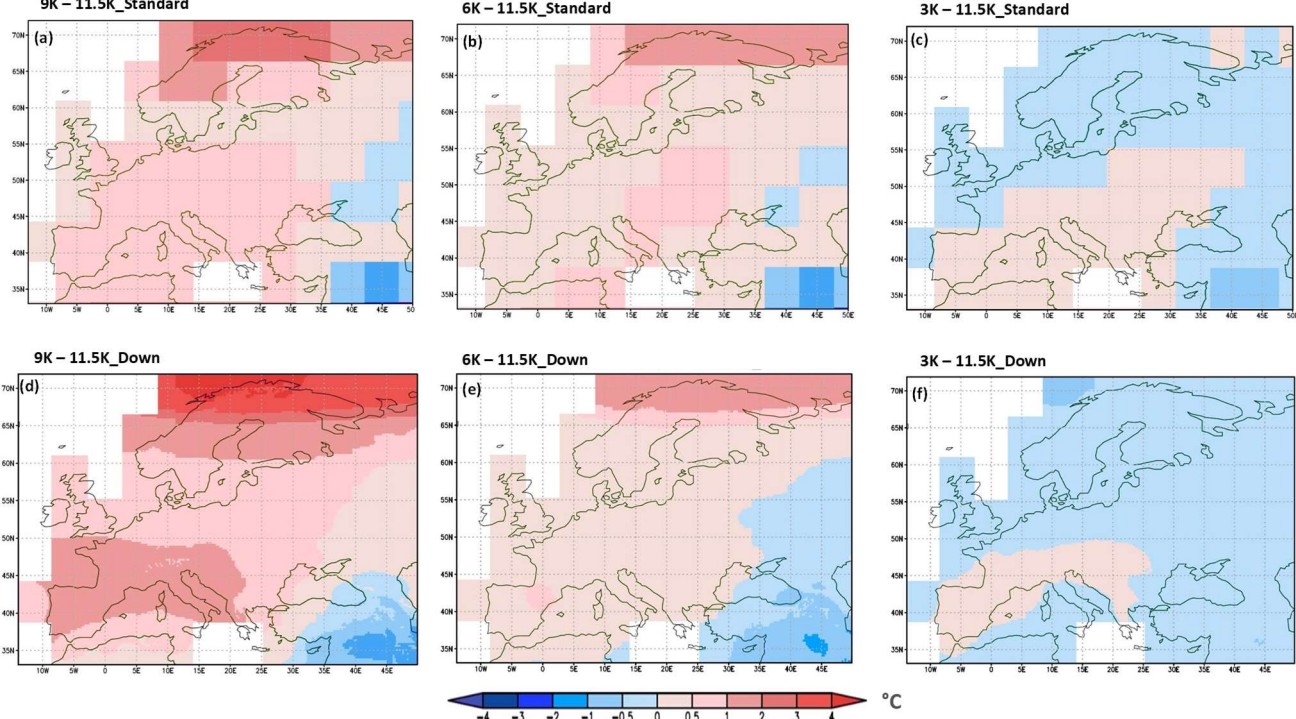

Figure 3. Simulated mean annual temperature anomalies (results minus the pre-industrial mean) showing spatial distribution in Europe for 9 kyr BP (a & d), 6 kyr BP (b, e) and 3 kyr BP (c, f) for 11.5K_Standard and 11.5K_Down.

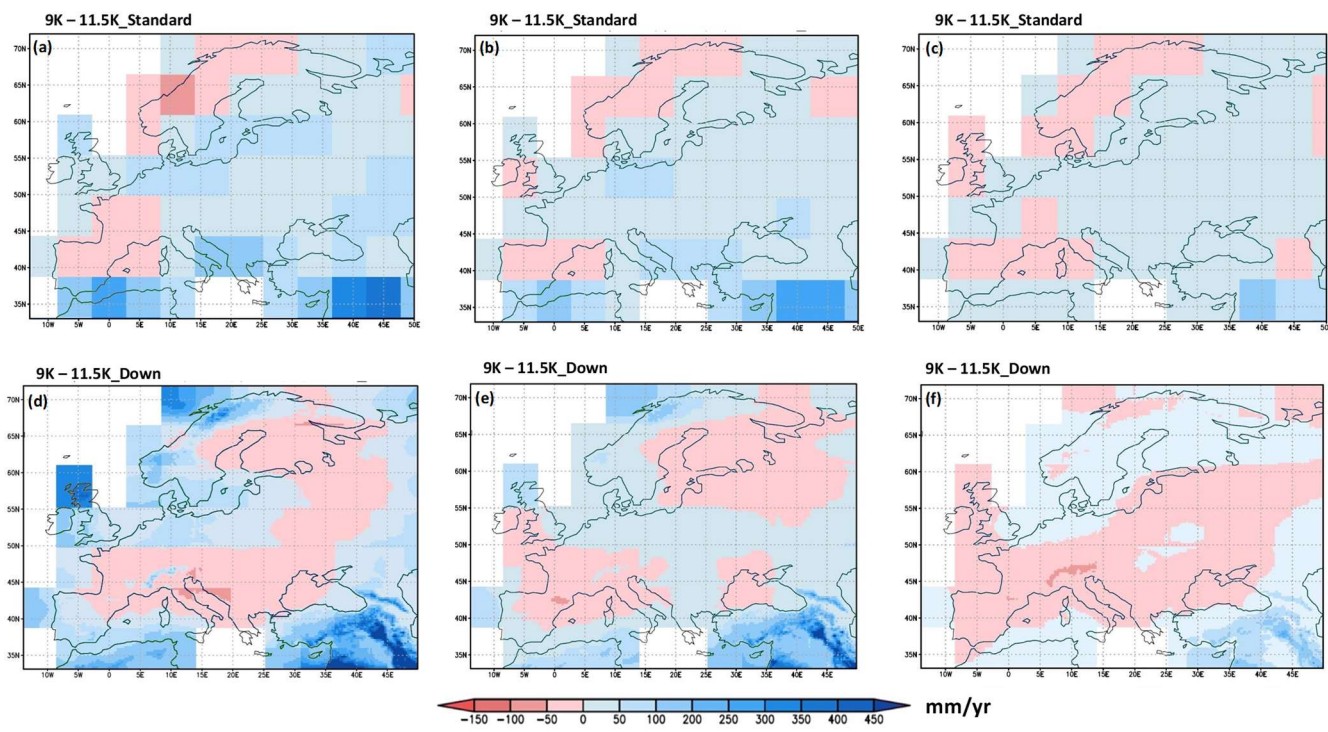


Figure 4. Simulated mean annual precipitation anomalies (in mm/yr, results minus the pre-industrial mean) showing spatial distribution in Europe for 9 kyr BP (a & d), 6 kyr BP (b & e) and 3 kyr BP (c & f) for 11.5K_Standard and 11.5K_Down.

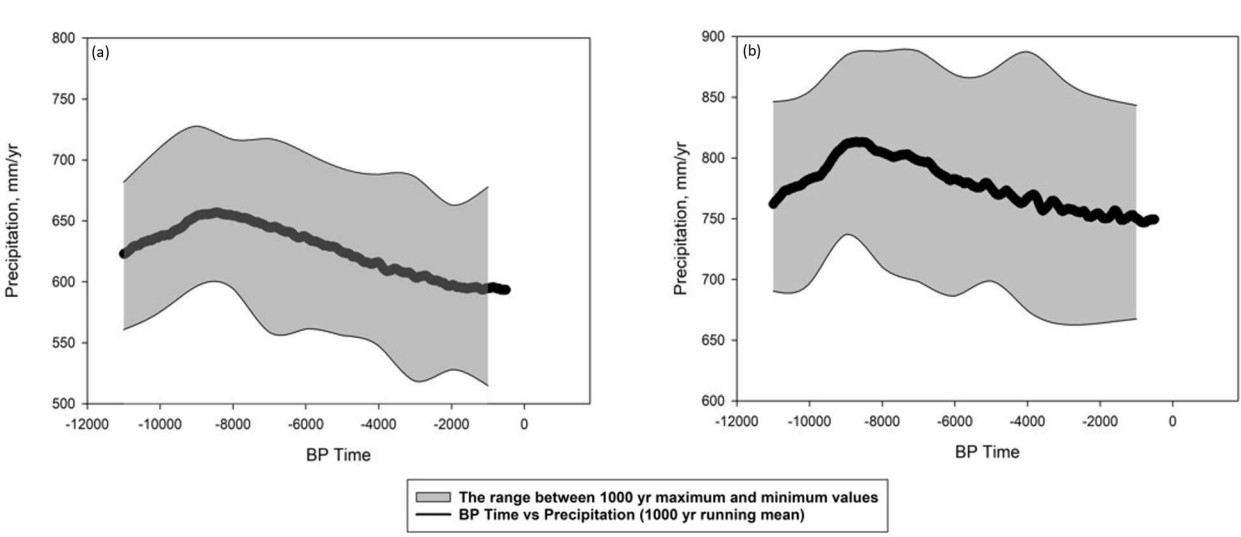

Figure 5. Annual Precipitation evolution in Europe during the Holocene for both 11.5K_Standard (a) and 11.5K_Down (b). Grey shaded areas represent the range between the maximum and minimum precipitation values in the transient simulations and the black curve shows the 1000-yr running mean of precipitation in mm/yr.

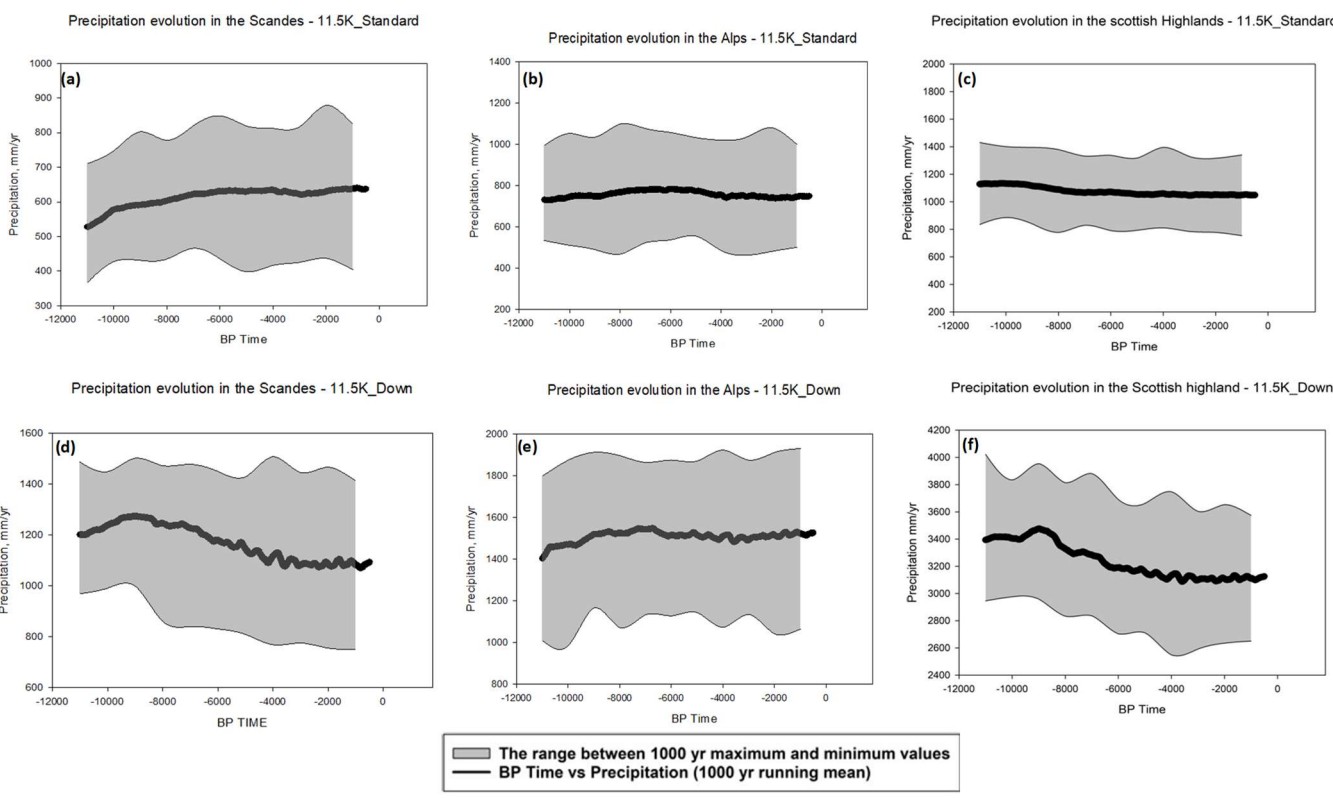

Figure 6. Regional annual precipitation evolution during the Holocene in some mountainous regions in Europe for both 11.5K_Standard and 11.5K_Down in the Scandes Mountains (a & d), Alps (b & e), Scottish Highlands (c & f). Grey shaded areas represent the range between the maximum and minimum values in the transient simulations and the black curve shows the 1000-yr running mean of precipitation in mm/yr.

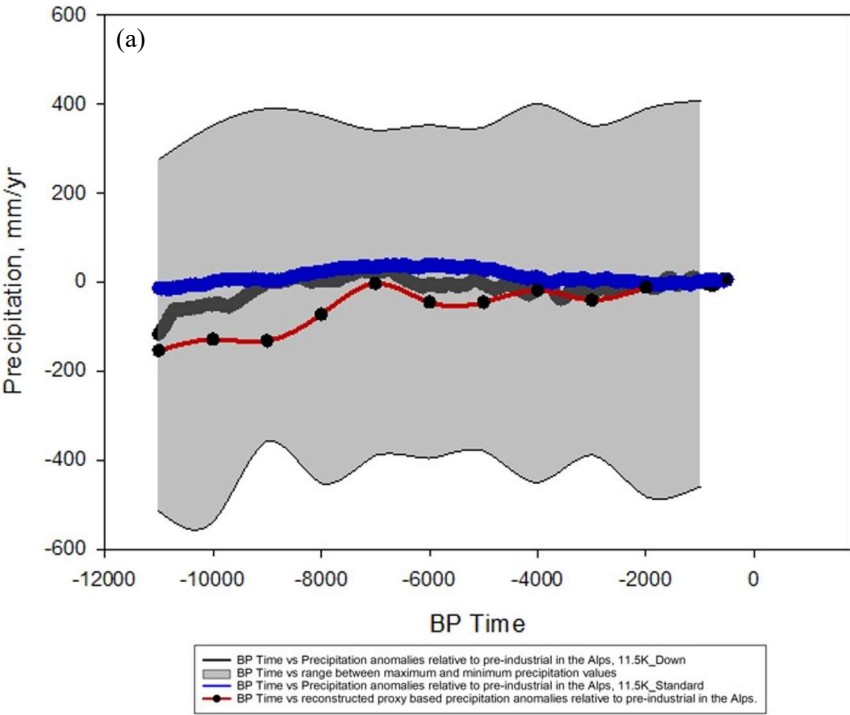

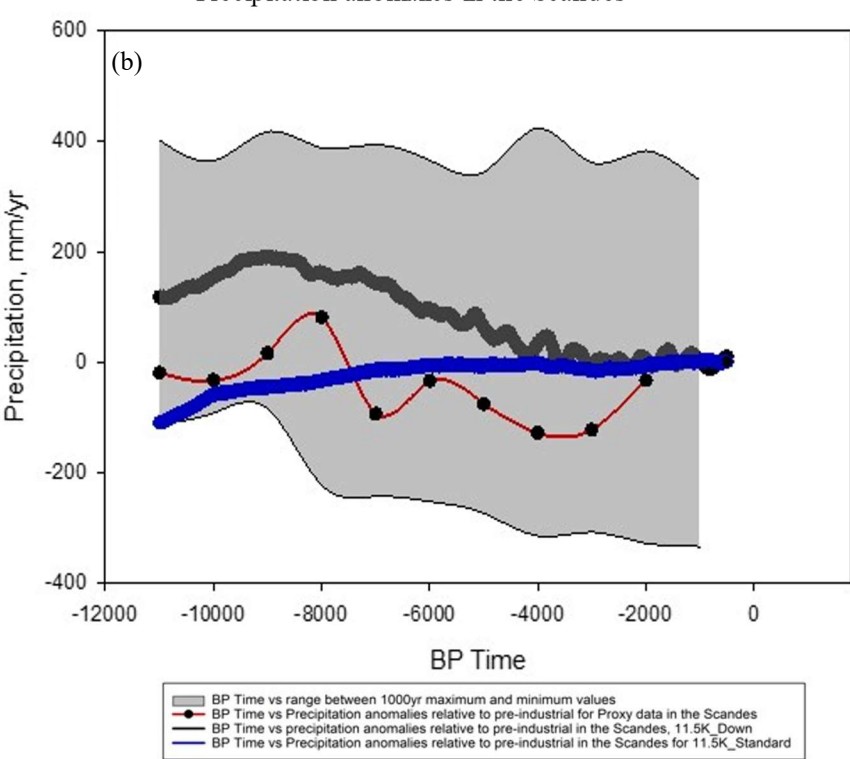

Figure 7. Comparison between simulated annual precipitation anomalies from iLOVECLIM and pollen-based reconstructions from (Mauri et al., 2015) in the Alps (a) and the Scandes mountains (b). The red line shows proxy reconstructions while the blue and black lines show the standard and downscaling simulations respectively. Grey shaded areas represent the range between the maximum and minimum precipitation values in the simulations and the black curve shows the 1000-yr running mean of precipitation anomalies relative to pre-industrial in mm/yr.

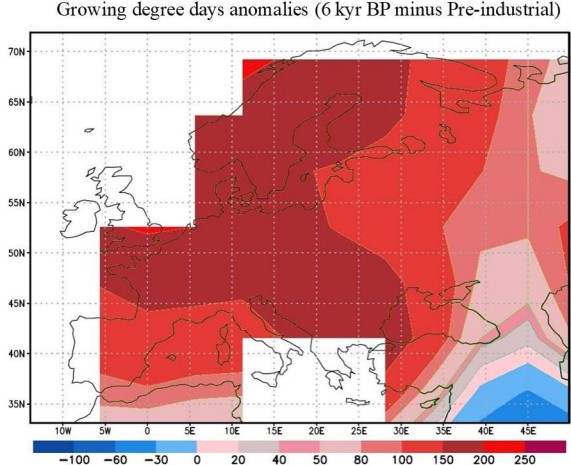

Figure 8. Simulated mid-Holocene growing degree days above 0ºC (GDD0) expressed as degree day anomalies (ºC) relative to the pre-industrial.

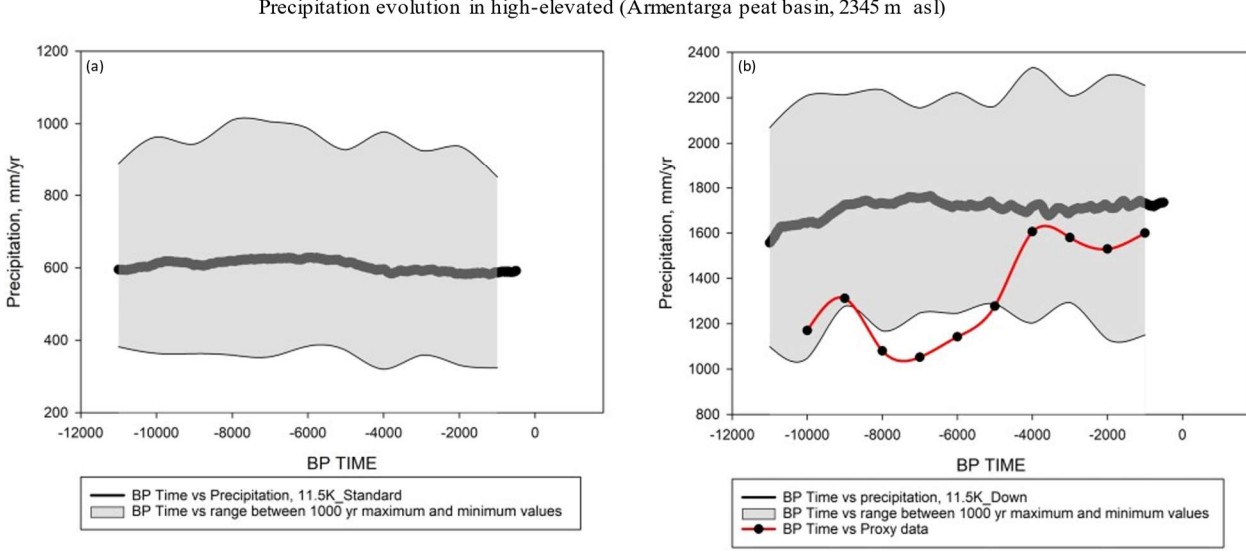

Figure 9. Simulated annual precipitation evolution with the standard version of the model (a) and interactive downscaling plotted with proxy-based reconstructions (Furlanetto et al., 2018), (b) at a high-elevated Armentarga peat (N 46° 2' 26.642"; E 9° 52' 44.263"), a site in the central Alps (2345 m asl). The black line shows the 1000-yr running mean of precipitation evolution during the Holocene in mm/yr while the light grey shaded areas represent the range between the maximum and minimum values in the simulations. The red line shows the proxy-based reconstructions at the site.