# Peer review of "Simulations of the Holocene Climate in Europe Using an Interactive Downscaling within the iLOVECLIM model (version 1.1)"

_Climate of the Past, 2022_

## Author Response (AR1)

**Reply to Reviewer #1**

*We appreciate the reviewer's thorough and constructive feedback on our article. We have responded to all his/her comments below (in italic font).*

**Summary**

The authors apply a "dynamical" downscaling technique to an orbital-only Holocene-long climate simulation and obtain a resolution of 0.25 degree over Europe. The downscaling is performed for temperature and precipitation. The authors were able to show that the downscaled precipitation matches precipitation and temperatures in mountain regions more realistically. In particular, the simulated trends of the downscaled data resemble reconstructions from different proxy archives.

*Reply: We thank the reviewer for a very good summary of what our aim and objective of this paper is about.*

**General**

The study touches a highly important topic in paleo climate, namely the mismatch between local climate proxy information and the rather coarsely resolved paleo climate modelling. The authors present an approach to bridge the gap of scales by a downscaling approach. Clearly the topic desires publication in the Climate of the past, but the current study shows a number of short comings. Besides some structural problems (see below) there is a lack in presenting the state of knowledge in the introduction as a large number of recent studies on dynamical downscaling for paleo climatic studies is missing. Then the results lack a clear discussion of seasonality differences in the proxy data. Also the method itself is to my opinion named wrongly: the authors called it dynamical downscaling and explain that the basic idea is to reproduce the model physics and NOT the dynamics. Due to my rather long list of comments (not sorted into major or minor) I recommend at least major revision of the manuscript.

*Reply: We thank the reviewer for pointing to us the gaps that are missing in our study. The reviewer gave some suggestions on recent studies related to dynamical downscaling. In our revised version, we have now provided a more thorough literature review on climate downscaling for paleoclimate studies. This literature review gives us the opportunity to explain better the originality of our approach and why we initially refer to our downscaling as «dynamical». We used "dynamical downscaling" in our case mostly because the downscaling is performed at each model time step, and it is consistent between the two grids (the precipitation at the coarse resolution interacts with the sub-grid). However, we understand that "dynamical downscaling" can be confusing for our approach since the regional climate model (RCM) community generally uses this terminology to refer to their approach. For this reason, we have changed the terminology in the revised version and now use "interactive downscaling".*

**Comments**

Title: The authors use only orbital and GHG forcing so I would not call it a Holocene Climate simulation, rather an orbital-GHG-only simulation for the Holocene period.

*Reply. We agree that short-term forcings (volcanic and solar) were not taken into consideration. We focus on the long-term multicentennial to millennial trends, so we consider it appropriate to have used forcings that are important at this time scale. Solar and volcanic forcings are known to be of particular importance for annual to decadal timescales. Simulations in the iLOVECLIM model show little impact of volcanic forcing in the Holocene transient simulations, but the impact is shown in the late Holocene (1.5kyr BP) (Bügelmayer et al. 2016). We prefer to keep "Holocene Climate simulation" to inform the reader we cover the entire Holocene.*

L 60 and the following paragraphs: There is growing literature on real dynamical downscaling using RCMs on the Paleo perspective so please make a reasonable review on the existing knowledge. Here is a collection of possible publications:

*Reply: We thank the reviewer for providing us with more suggestions about previous studies related to dynamical downscaling using RCMs and we also agree with him/her that our study needs more examples to motivate our work. In particular to highlight the pros and cons of the different approaches. In our revised version, we have included the suggested literature in the introduction.*

L61: There are several approaches to statistical downscaling simulation. An example is Latombe et al. 2018, but there are several more publications so I encourage the authors to make a lit. search to add it to the introduction

*Reply: We agree that, similarly to RCM downscaling approaches, statistical downscaling can also provide useful information for paleoclimate studies. We have now included such studies in the introduction.*

L65: The publication Feser et al. is not dealing with paleo research questions so why is it cited?

*Reply: The publication of Feser et al. is about how Regional climate models add value to Global model data. We now only cite the paper where we explain how RCMs are useful for dynamical downscaling because their output data may be processed to produce higher resolution atmospheric fields, allowing for the modelling of small-scale processes and a more complete description of features (such as mountain ranges, coastal zones, etc).*

L80-85: Here is another place to add model publication of real dynamical downscaling.

*Reply: We have done so in the revised manuscript.*

L95: Units are not in italic

*Reply: This is corrected.*

L100 and following: There are several more studies doing Holocene long simulation. A recent one is Bader et al. but please check again the literature. There are also approaches which first use a coarse resolved AO GCM and then a high resolved A GCM forced by the SST and sea ice distributions, please check Merz et al. and Hofer et al. publications.

Bader, J., et al. 2020: Global temperature modes shed light on the Holocene temperature conundrum. Nat. Commun., 11, https://doi.org/10.1038/s41467-020-18478-6.

Hofer, D., et al. 2012: The impact of different glacial boundary conditions on atmospheric dynamics and precipitation in the North Atlantic region, Climate of the Past, 8, 935-949

Hofer, D., et al. , 2012: Simulated winter circulation types in the North Atlantic and European region for preindustrial and glacial conditions, Geophys. Res. Lett., 39, L15805

Merz, et al. 2013: Greenland accumulation and its connection to the large-scale atmospheric circulation in ERA-Interim and paleo-climate simulations, Climate of the Past, **9**, 2433-2450

*Reply: Thanks for these important publications. They are now part of the revised introduction.*

Introduction in general: A discussion on the so-called 'Holocene temperature conundrum' is missing. Please check Bader et al 2020 and Liu et al. 2014.

Liu, Z., et al. 2014: The Holocene temperature conundrum. Proc. Natl. Acad. Sci., 111, 3501–3505, https://doi.org/10.1073/pnas.1407229111.

*Reply: This is a very good point; We thank the reviewer for providing us very good literature suggestions. We have included the literatures proposed by the reviewer in the revised manuscript.*

L125: Please change in the caption to "extent".

*Reply: This is corrected.*

L 129: Why do you go from section 2 to subsubsection 2.1.1? This makes no sense. This is happening several times. There are also sections where there is only one subsection which is again awkward. Please correct the structure according to the rules of the journal

*Reply: The structuring of the paper is now done in the revised manuscript by following the Journal's rules.*

L140: The ECBilt model is a quasi-geostrophic model so the most important mode of variability in the climate system ENSO is not included in the model by definition. So how does this shortcoming impact your results knowing that ENSO has an influence on the Europe?

*Reply: The impact of ENSO is expected to be minor in our simulations because we are investigating long (multi-millennial) trends while ENSO's impact is normally on short timescale.*

Section 2.1.2: The authors do NOT apply a dynamical downscaling as they correctly say that the only try to reproduce the model physics and not the dynamics so it is awkward to call the method "dynamical downscaling. This is an important point as real dynamical downscaling implies the application of a regional climate model with incudes dynamics.  So I recommend that the authors change the wording in the entire manuscript.

*Reply: We used "dynamical downscaling" in our case mostly because of two reasons. First, the downscaling is performed at each model time step during run time. Second, there is a two-way coupling between the coarse grid and the sub-grid which ensures consistency (the precipitation at the coarse resolution is the sum of the sub-grid precipitation). As such, there is a strong difference with standard offline downscaling techniques. We thought that the term "dynamical" is good to convey these two ideas. As mentioned previously, we have changed the terminology to "interactive downscaling" in the revised version and explained better the originality of our downscaling procedure.*

L158-59: If I understand this correctly the method conserves the precipitation amount so that I would average over the same area as the coarse grid I would obtain the same precipitation also in the fine grid. If this is correct I do not understand why precipitation is different in Figure 5and e.g. in Fig 4 if we look at the grid point over Scotland. So either the figure is wrong or the description of the methods is incorrect.

*Reply: We do not fully understand this comment because Figure 5 shows the precipitation trend for the whole of Europe while figure 4 shows the simulated precipitation anomaly in spatial distribution in Europe relative to the preindustrial. The figures 4a and 4d are different because we have two different experiments: an experiment with downscaling and one without the downscaling. This implies that the evolution of climate in the two experiments is different, as seen in Figure 5. The experiment with downscaling modifies the amount of precipitation with respect to the standard model given that it accounts for the sub grid orography and the non-linear relationship between temperature and humidity.*

L159-164: Well isn't this logical as the method does not include a dynamical part (only physics is changed) one would expect that it is not able to change the biased large scale atm. circulation.

*Reply: Yes, we agree with the reviewer here on the fact that we cannot correct the biases related to atmospheric circulation. However, the downscaling changes the whole pattern of precipitation, generally producing higher precipitation over mountain ranges and as a result some drying over the continental plains. In this way it could have corrected some of the biases (or equally worsen them, depending on the bias considered).*

Section 2.1.3: To my understanding I would not call this a Holocene simulation as important external forcing agents are missing, i.e., solar forcing and volcanic eruptions. It is clear that the authors cannot rerun the simulations using all forcings so I suggest to make it clear that the authors performed an orbital-GHG-only simulation for the Holocene period. So name the simulation always "an orbital-GHG-only simulation for the Holocene period." Just for curiosity why do you only use orbital and GHG forcing and not include the other two?

*Reply: As replied earlier, we focus on the long-term multicentennial to millennial trends, so we consider it appropriate to have used forcings that are important at this time scale. Solar and volcanic forcings are known to be of particular importance for annual to decadal timescales. Simulations in the iLOVECLIM model show little impact of volcanic forcing in the Holocene transient simulations, but the model responds to volcanic forcings in the late Holocene (1.5 kyr BP) (*Bügelmayer et al. 2016)*.*

L245: Why do the authors compare their results to PMIP2 and not to PMIP3 or 4? There are newer studies e.g. Liu et al. 2014, Russo et al. 2022 and PMIP4 studies.

*Reply: We have now compared our results with the latest PMIP4 papers in the revised version.*

L253-54: please change to "Overall the native grid (T21/11.5_Standard) is still seen on the 11.5K_Down model results in many regions for all times slices. "

*Reply: This is corrected, thank you.*

Fig.4: The downscaled data looks weird, e.g. in panel d we see at 50N a clear boundary with a change from +100 mm/yr to -100 mm/yr with no gradient in between. This makes no sense

*Reply: The downscaling still uses the T21 grid, creating some unrealistic gradients. This is a limitation to our methodology.*

3.2.1 is the only subsubsection which makes no sense.

*Reply: We have corrected this.*

4.1.1 The authors compare their results to proxy data which is good. Still I miss a clear discussion on the seasonality of the proxy data which might play an important role in interpreting the proxy data especially the trends. E.g. tree rings and pollen data are biased to the growing season but these data are compared to yearly means of the simulation. Check out the Bader et al 2020 publication on this.

*Reply: Seasonality is important,and we have made a short discussion of this in the introduction and discussion in our revised manuscript.*

L442: Brayshaw et al. does not simulate the entire Holocene. He rather simulated time slices distributed during the Holocene. So it is not a transient simulation he performed. Please be more specific about this.

*Reply: We thank the reviewer for this comment. We have reworded to explain to the reader that the work of Brayshaw et al was not transient simulations.*

L475 and paragraph: Again only PMIP2 is used, why not using the updates of PMIP3 and 4 ?

*Reply: We have added the updates of PMIP4 to our revised manuscript.*

4.1.2 This subsection is rather short compared to the 4.1.1 so just merge it to one section 4 Discussion

*Reply: We have merged them together in the revised version.*

L485-87: I think there is a caveat which makes the data not so useful as the authors think as the coarse grid sometimes remains preserved in the downscaled data leading to boundaries (see Fig. 5). I think the authors need to be more cautious about this and not overrate their results.

*Reply: We thank the reviewer for this comment, there is a limitation to the downscaling methodology. Regardless of this limitation, the downscaling data is at least able to show more spatial details which is lacking in the course resolution. Yes, we agree with the reviewer to be*

*cautions in exaggerating our results. We have now revised in the manuscript with the choice of words.*

Reference list contains a lot of errors please correct them.

*Reply: We have checked them .*

The quality of the figures is bad please use at least 300 dpi.

*Reply: We have checked the quality of the figures and its above 300 dpi.*

**Reply to reviewer #2**

*We appreciate the reviewer's thorough and constructive feedback on our article. We have responded to all his/her comments below (in italic font).*

**Summary**

The authors present a Holocene climate model simulation for Europe at a high-spatial resolution using dynamical (as opposed to statistical) downscaling. This regional model simulation uses inputs from the global iLOVECLIM EMIC. The authors compare this simulation with climate reconstructions and conclude that their higher-resolution model better matches the data than the lower-resolution EMIC. The main innovation is applying this method to a transient simulation that encompasses the whole Holocene and not just time-slices. I am with the authors here, in that I think that scaling down model results to the spatial scale of the proxy data is a potentially good idea, and one well worth investigating.

*Reply: We thank the reviewer for summarising the motive behind our studies in a concise and clear understanding to readers.*

**General comments**

The premise of the paper is interesting and is clearly appropriate for publication in Climate of the Past. The main problem is the nature of the evaluation which is qualitative and anecdotal rather than being rigorously quantitative. The authors really need to include a better designed evaluation process where proxy reconstructions are compared with the model results on a site/record basis, or for a region, in the case of gridded reconstructions. Improvements in the evaluation should also be extend to comparisons between the EMIC and downscaled modeling, for instance plotting both in the time-series plots. It would also be useful to compare the iLOVECLIM EMIC that is used with other models (eg PMIP3 GCM's) to see what particular biases this particular model has over the study region.

The authors also appear to be a bit loose with their commentary. When referring to temperatures and temperature changes it is essential to state what temperature variable it is that they are referring to, for instance whether it is mean annual, winter, summer or some other aspect of temperature. This appears to be a source of confusion throughout, and particularly when citing other studies as supporting evidence. Similarly, in the discussion of proxy data, the authors need to distinguish between studies that provide evidence from single sites, and those that provide evidence from many hundreds of sites, since they are not equivalent. Also, it is important to understand the different studies cited; In Southern Europe, Bartlein et al is essentially a synthesis of the data of Wu et al 2007 and Davis et al 2003, Brewer et al 2007 takes data from Davis et al 2003, and Mauri et al 2014/15 is an improved version of Davis et al 2003 and Brewer et al 2007, both of which it supersedes. All of these latter studies use gridded data, not site data, although the gridding itself is based on site data.

I will also just add here the importance of isostatic uplift over Scandinavia during the Holocene, which in some areas has been substantial (100m+). Data-model comparisons over Scandinavia should be treated with caution where the model uses modern topography but is compared with proxy reconstructions that were much lower in the early Holocene

*Reply: We thank the reviewer for this comment. We would like to note that the main intention of our paper is to evaluate our downscaling result against the standard version of the model*

*which has a much lower spatial resolution. It was not our intention to make a full model-data comparison here, but we agree that it is very valuable to compare our high-resolution results with proxy-based reconstructions for specific mountainous regions where the potential of our downscaling method is especially clear. In this revised version, we have compared our results (both the standard version and the downscaling) with Mauri et al., 2015 proxy data as suggested by the reviewer and we have clarified in the discussion about the type of proxy data we have compared our results with. In the discussion part we have compared some GCM results from the PMIP4 with our model results, In the revised version, we have now explained clearly the temperature terms (such as annual and seasonal temperature when comparing with our work to avoid confusing the readers.*

**Detailed comments**

67-69 (and 160-164) This is a critical point.. the regional downscaling cannot correct major errors and biases in the global model simulation, including atmospheric dynamics which have been suggested as the source of much of the data-model discrepancy over Europe during the Holocene (Mauri et al 2014). The authors should note how the iLOVECLIM model generally compares with other global model simulations (eg PMIP3), for instance if it is generally cooler/warmer or wetter/drier than average, or comparable.

*Reply: yes we are aware of this.*

77-78 The main 'sensor' area of a proxy-based climate reconstruction is rarely greater than ~20 km radius for pollen and can be as small as a couple of hectares for lake-based proxies such as chironomids. It therefore makes sense to undertake data-model comparisons at comparable spatial scales (see my opening comments about improving the data-model comparison.

*Reply: Yes, a very important point here, data-model comparison is hindered by the different characteristics of each dataset at small spatial scales, model output is less reliable. To avoid unbiased comparison between our model results and the proxy data, it is important we consider the spatial scales. We have taken note of this in our revised version of the manuscript and plotted our results with the Mauri et al data.*

101-106 This is misleading. While Brewer et al does suggest that climate models can simulate cooler temperatures over Southern Europe and the Mediterranean during the mid-Holocene, this is only in WINTER and the signal is very weak. In contrast, reconstructions of SUMMER temperatures are much cooler than the models, which all show warmer summer temperatures (ie not even the same sign). This is not discussed by Brewer et al, but is clearly shown in the more recent reconstruction by Mauri et al 2014. Both Brewer et al (who uses the data from Davis et al 2003) and Mauri et al use pollen data, but the problem with cooler summer temperatures is also shown in SST reconstructions for the Mediterranean, as shown in Hessler et al 2014 (doi:10.5194/cp-10-2237-2014) figure 4. In fact, the data-model discrepancy shown in Mauri et al 2014 is a very good justification for the authors to have undertaken their study

*Reply: Thanks for the comment, we have rephrased this paragraph and we have been more specific in the revised version of the manuscript. We did not use the Mauri et al 2014 because our paper is mainly focus on annual transient simulations in the Holocene whiles the Mauri et al 2014 was generally on summer and winter.*

119-121 I don't really understand why the authors have chosen specific areas (and variables, eg precipitation) where they then say they don't actually have proxy records, if their stated aim is to make comparisons with proxy records. It seems that they did the model analysis first, and then looked for proxy records afterwards.

*Reply: With this study we provide the first transient simulations of the last 12k with interactive downscaling at 0.25° resolution over Europe. For this reason, we first wanted to assess the major changes induced by the downscaling with respect to the standard non-downscaled version of the model. The downscaling has a major impact over mountainous Region, so we focus our analysis there.*

194-195 The authors use a pre-industrial climate baseline to calculate anomalies to compare with climate reconstructions. I hope that the authors are aware that anomalies shown in almost all proxy-based reconstructions are based on a modern baseline (apart from for instance Davis et al 2003 used in Brewer et al 2007, and Mauri et al 2014/5 that use a pre-industrial baseline of ~1850)

*Reply: Yes, we are aware of this, but we thank the reviewer for bringing our attention to this.*

234+ Please be very careful, do not use the unspecified term 'temperature/s'. Please always state if this is annual, seasonal (JJA, DJF) etc. The authors appear to be conflating winter (Brewer et al 2007) and annual temperatures (Wu et al 2007) in the data, while the temperatures you are referring to in the model results are unspecified.

*Reply: This has been resolved in the revised version.*

246 Again this is misleading. Brewer et al only considered winter temperatures. Better to refer to Mauri et al 2014/15 which is a more recent and more comprehensive study that includes summer winter and annual temperatures (and precipitation).

*Reply: We thank the reviewer for this comment. This has been done in the revised version,*

249 Wu et al uses an inverse modelling method, so represents a very different pollen-climate reconstruction to the MAT method used by Davis et al (in Brewer et al) and Mauri et al, although both show the similar results (see Davis 2017 https://doi.org/10.22498/pages.25.3.16). Note also that Wu et al is for individual sites, while Brewer et al use a gridded reconstruction where the site data has been interpolated onto a 1 degree spatial grid. There are also considerably more sites in Brewer et al than in the Wu et al reconstruction, while the sites used in Wu et al are poorly dated, use truncated taxa assemblages (a lot of data is from Huntley and Birks 1983), and have large uncertainties

*Reply: Thank you for this informative comment. We have taken this into consideration in our revised version.*

258 Fig.3 What aspect of temperature is the figure showing? Mean annual, summer, winter etc ? please specify

*Reply: This represents annual mean temperature anomaly. It has been corrected in the revised version.*

264+ Again, as with temperature, please specify what aspect of precipitation you are talking about.. I presume its mean annual (units are in mm/yr), but please state this clearly at the start

*Reply: Thanks for the comment, this is annual mean precipitation, so this has been clarified in the revised version.*

280-288 This is interesting, if not surprising. A question arising from this would be if the downscaling simply spatially redistributes the average precipitation of the EMIC grid box, or does it potentially increase/decrease the average precipitation that would occur in the EMIC grid box?

*Reply: The downscaling does not simply redistribute the precipitation, as can be seen by a comparison for three mountainous regions in Figure 6. This clearly shows that with downscaling the precipitation is higher.*

323 Fig. 5. & 371 Fig. 6, Section 4.1.1 Why have the authors chosen not to do a data-model comparison for the precipitation time-series for the Alps, Scandes etc? This would be very straight forward using the data from Mauri et al 2015 which is freely available. Also, why not show the EMIC result for the same spatial areas? This would illustrate the difference between the EMIC and the downscaling (and data)?

*Reply: we have done this for the Alps and the Scandes using the Mauri et al 2015 data in the revised version.*

399-413 The Furlanetto et al paper consists of only one site from the Alps, the Mauri et al analysis consists of hundreds of sites from the Alps (this has been gridded, but there is also the underlying site data that could be used). Why did you pick only this particular study? In any case, it would be useful to plot the Furlanetto et al precipitation reconstruction against the model result (both high and low resolution) so that the reader can see for themselves. It is also notable that the authors identify the strong spatial variance of the precipitation signal, but compare this single proxy site with the average precipitation of the entire Alps. Surely it makes more sense to compare the proxy record with the nearest point in the model grid?

*Reply. Our downscaled simulation provide precipitation results at a relatively high spatial resolution. To evaluate this kind of model result, especially when evaluating against a low resolution model version, it is of crucial importance to use proxy-based records that capture this same high spatial resolution. This is the reason why we use the Furlanetto et al. paper. It is not easy to find other proxy data with such high resolution. The Mauri et al. dataset is very valuable for model evaluation, but it presents gridded precipitation at 1° resolution, which in our view does not represent a high enough resolution to represent specific conditions in the Alps or the other mountainous regions we focus on. Nevertheless, we have followed the suggestion of the reviewer and included a direct comparison with the reconstruction of Mauri et al. (2015) for the Alps and the Scandes. As mentioned before, the main purpose of our study is to compare the downscaled results with the standard low-resolution results. We take the entire Alps because a smaller region would not make much sense for the low-resolution model. We compare to Furlanetto et al. to make the point that the high-resolution version provides not only more spatial detail, it also generates precipitation values that are much more reasonable than the anomalously low precipitation values of the low resolution version. The Furlanetto et al. record is thus indicatively used to make this point. We thus do not claim that this record is*

*representative for the entire Alps, and we therefore prefer not to plot the Furlanetto results directly against our model results.*

.

415-429 Again, as with the Alps, it would be better to directly show the proxy reconstructions plotted against the model result, and even better to show this at the model grid point closest to the site (or interpolated to the site location). If the authors really want to compare using the entire Scandes region, then at least compare against the same area using the Mauri et al 2015 gridded data, since this is designed to avoid spatial sampling bias associated with simply averaging site records together. It is also important to note that (presumably) the model uses modern topography and does not take into account the substantial changes in elevation that has occurred during the Holocene due to isostatic uplift. This is important when comparing with proxy records that actually include this isostatic change (See Mauri et al 2015)

*Reply: Yes, a very important comment and we have plotted our data with those from the Mauri et al. (2015) paper*

448-465 The authors are conflating proxy reconstructions here across all kinds of spatial scales, some from individual sites, some based on the synthesis of large numbers of sites to represent individual regions, and some where the site records are projected onto a spatial grid. Again, it would be better if at least some of these records were compared explicitly with the model (ie one plotted over the other) rather than resorting to a rather vague 'one thing looks like another thing' statement, which is open to interpretation. This is particularly important because the ability to compare model and proxy record at the scale of the proxy site is supposed to be one of the main advantages of the model downscaling that the authors are proposing. Including the results of the EMIC in the same way would also help demonstrate this

*Reply: We agree with the reviewer that the ability to compare our model results and data at the scale of the proxy site should be treated as a high importance.*

475 Southern Europe in the mid-Holocene in the PMIP2 simulations is warmer not cooler (only winter is a little cooler in the far east) and with little change in precipitation. This is shown in detail in Mauri et al 2014.

*Reply: We have clarified this paragraph in the revised manuscript.*

475-478 Please do not write about climate model results as if they are some kind of reality. For instance "we can infer from their work that southern Europe was wetter and cooler." Should read something like "eg we can infer from their work that southern Europe was wetter and cooler in PMIP2 model simulations."

*Reply: Thanks for the suggestion, this has been corrected in the revised version.*

475-482 The PMIP2 results encompass a large number of different models, each sometimes showing quite different results. Are you talking about individual PMIP2 models, the ensemble mean or something else? Please be more specific. Also, Braconnot et al 2007 does not show any detail for Southern Europe or the Mediterranean (but is shown in detail in Mauri et al 2014) so I am not sure how the authors are making their comparison unless they have been plotting

the data separately (it would be great to actually show this). In any case PMIP2 has been superseded by PMIP3.

*Reply: We have discussed our results with the PMIP4 now.*

485 Grammar needs correcting: "in the change pattern for"

*Reply: We have revised as suggested.*

485-495 What about temperature lapse rates? Changes in temperature lapse rates as a result of the downscaling will also lead to change in temperatures at different altitudes, perhaps better reflecting the proxy data.

*Reply: The vertical lapse rate in temperature is computed in our model and shows a representative of the free-atmosphere temperature variations. So, we agree with the reviewer here that the lapse rate had an effect with temperature changes with elevation and could better represent the poxy data.*

496+ Conclusions- see my opening comments. The study needs a more rigorous approach to the data-model and model-model comparison.

*Reply: We thank the reviewer for taking time to make suggestions on ways to get an improved version of our manuscript. We have worked on his/her major suggestions which is more based on the evaluation of our results with proxy data and other model data.*

---

## Author Response (AR2)

**Reply to Report # 1: Anonymous Referee # 2**

Summary: In this revised version of the manuscript, the authors have made some improvements according to my recommendations but have also neglected to correct other problems. The manuscript includes lots of small errors with the English and careless typos, seemingly worse than the first draft.

*Reply: We have done all the recommendations suggested by the reviewer and corrected the small errors and English in the revised manuscript.*

English. There are many small but nevertheless significant typographic and grammatical errors in the English. Perhaps your co-authors could help you here. I highlight only a selection but there are many more.

*Reply: We thank the reviewer for pointing to us the typos and grammar errors in the manuscript, we have made corrections to these errors in the revised manuscript.*

15. 'Reconstructions' not 'reconstruction'

*Reply: We have corrected this.*

151. 'and FOR almost all climate models'

*Reply: This has been corrected.*

152. 'warming IN SUMMER in the Mediterranean' Models response is tightly connected to insolation change at the mid-Holocene; summer insolation was higher and winter insolation was lower and temperature reflect this, including over the Mediterranean region.

*Reply. We thank the reviewer for these suggestions, we have now amended this sentence in the new version.*

153-156. See my original comments about making false equivalence's between continental scale reconstructions based on 100's of sites and averaged over 1000's of km2, and individual sites that reflect only very local conditions of only a few km2. Even though temperatures may show cooling on average across the Mediterranean, it does not mean that warming did not occur in some areas, that's how 'averages' work. The paper by Samartin et al is disingenuous in its approach. The glaciers and chironomid records cited by Samartin et al are in central Italy, where pollen records also show a mid-Holocene warming entirely in agreement with the chironomid and glacier records. For instance, see warmest month temperatures from the closest pollen record (originally by Marscicek et al) in the Climate12k compilation https://lipdverse.org/Temp12k/current_version/Ospitale.Watson.1996.html, and compare it with the Samartin et al. July chironomid record just a few km's away https://lipdverse.org/Temp12k/current_version/LagoVerdarolo.Samartin.2011.html). The Alkenone SST records that Samartin et al. mention (I presume these are the 'marine cores' that Arthur et al mention in line 155) represent annual temperatures, not summer temperature records comparable with the chironomid records. Alkenone records have been shown to be subject to bias in the Mediterranean caused by seasonal changes in productivity (eg Grauel et al 2013 doi: 10.1016/j.quascirev.2013.05.007). Actual summer SST records based on forams show an average cooling across sites in the Mediterranean (see Hessler et al 2014 doi:10.5194/cp-10-2237-2014, Fig. 4).

*Reply: We thank the reviewer for the broad explanation of the data comparison here. We have now amended this section to make it clear to the reader.*

Fig.2. 0 BP is AD 1950, but GHG does not show appropriate levels for 1950. If GHG are pre-industrial then the age scale needs to be adjusted accordingly.

*Reply: We have corrected the figure and included a revised version.*

300 (and in the rest of the manuscript). Almost every time you use the word 'anomaly', you should in fact be using the word 'anomalies'.

*Reply: We have made the corrections in the entire manuscript.*

303. 'SPATIAL pattern'.

*Reply: This is done*

477-478. Please remove 'likely' and replace with 'could be' or something similar. Bader et al. 2020 provide at best only circumstantial evidence that proxies could be biased towards the growing season. They rely almost entirely on correlation to demonstrate causation, and only for selected sites/records. Their argument would quickly collapse if they had used a broader selection of records that would have shown a wider variety of temperature trends for the same latitude, some positive, some negative (eg see the wide variety of Holocene temperature trends in https://lipdverse.org/Temp12k/current_version). The more convincing way to investigate this problem would have been to use a process-based model of the proxy, then change the seasonal insolation to see what the effect would be. In fact, this has already been done quite extensively for pollen using vegetation models run in inverse mode (see inverse modelling in Chevalier et al 2020 doi:10.1016/j.earscirev.2020.103384), and no evidence has emerged that seasonal bias is a factor in pollen reconstructions. For instance, comparisons of reconstructions based on inverse modelling (using a vegetation model) and modern analogue methods (using modern surface sample calibration datasets) show no discernable difference. For example, for the mid-Holocene see Davis 2017 Doi:10.22498/pages.25.3.161, Fig.1, and for the LGM see Davis 2022 Doi:10.5194/cp-2022-59, Fig.7.

*Reply: We thank the reviewer for an extensive explanation of seasonal bias, we have amended this paragraph as suggested by the reviewer.*

480-485: I mentioned this in my first review, and the authors don't seem to have understood the problem. It makes no sense to compare an area-averaged record for the whole Alps with a single site in the Italian Alps. Precipitation varies considerably across the Alps at a regional and local scale. The authors need to compare like with like using either the closest grid point, the closest grid box or (best of all) an interpolation to the exact x, y, z location of the site. Presumably this is where the advantages of the higher resolution regional model simulation should be clear. At the very least the regional model should show a higher precipitation value because the larger grid box of the GCM will have a lower average altitude. This would actually help demonstrate the usefulness of the higher resolution model. If the authors do include this comparison, then I would ask for them to show the reconstruction by Furlanetto et al and the model simulations as a time-series figure, even in the supplementary.

*Reply: We thank the reviewer for this suggestion, we have now plotted our high-resolution results to the exact location of the site used by Furlanetto et al paper in the Alps and our high-resolution model show higher precipitation values as the reviewer pointed out above.*

495-498. I also mention in my first review the importance of post-glacial uplift in the Scandes mountains when making comparisons between data and models. The authors still make no mention of post glacial uplift in the revised manuscript. The precipitation reconstruction in Mauri et al does not correct for uplift, only temperature is corrected.

*Reply: We did not correct for post glacial uplift for precipitation in the Scandes mountains and we have now stated this in the discussion.*

498. 'DOES not reflect the underlying topography'? although I don't know what the authors are trying to say here.

*Reply: We have corrected this.*

526. Missing bracket after 'Peyron et al 2017)'.

*Reply: This is now corrected.*

563-581. The English contains many simple errors. 563. This sentence is incomplete. 565. 'precipitation ANOMALIES' 567. Same. 575. 'DOES not capture. 576. 'and much wetter' 578. 'the PERSISTANT mismatch' or similar, but not incessant.

*Reply: We thank the reviewer for bringing these to our attention, we have now corrected the errors in the entire manuscript.*

616. '(For example..' should be '(for example..'

*Reply: This is corrected.*

**Reply to Report # 2: Anonymous Referee # 1**

Overall, some of my concerns have been treated with caution but there still a couple of issues which needs to be solved.

*Reply: We thank the reviewer for taking time to review our work. We have resolved the issues in our revised version.*

L22-23 The last sentence of the abstracts reads bad, please revise.

*Reply: We have done this correction.*

L76 missing point and I would avoid the line break here.

*Reply: This has been done.*

L121-125 This paragraph seems to be at the wrong place. I suggest to merge it with paragraph from L171-181.

*Reply: This has been done as suggested by the reviewer.*

All figures have still a bad resolution it is maybe an issue of the uploading the files and how the files are handled by climate of the past. please make sure that in the final version the resolution is sufficient.

*Reply: The issue with the resolution has been resolved as well as the technical challenges due to the upload of files by Climate of the Past and the figures now have good resolution. The final version will have separate .jpg files above 600 dpi for all the figures which will be attached to the manuscript.*

Fig. 1: The color scale does not fit to the shading used in the figure. There use just use the standard colorscale of ncview which is not appropriate for a publication.

*Reply: We have resolved this.*

L194: please change to "2 Model, Simulations and Methods"

*Reply: This has been done as suggested.*

Fig. 2: The aspect ratio of the figure is changed so please avoid this. There is a strange A as head of the figure why? I suggest to produce an own plot as the authors performed the simulations so they should have the forcing files.

*Reply: We have done this as suggested by the reviewer.*

L301: It still remains unclear how the authors had calculated the anomaly, is it the different between 9kyr and preindustrial or vice versa or the difference between 9kyr and the long term mean of the simulations? The sentence in L261-264 is highly unclear

*Reply: The anomaly is calculated as the difference between 9 kyr BP and pre-industrial (that is, 9 kyr BP minus Pre-industrial mean). The sentence has been clarified in the revised version.*

Table 1 a bracket is missing "(downscaling)"

*Reply: This has been corrected.*

L309: "warm with an annual temperature"

*Reply: This is now corrected.*

L311: "part, which had"

*Reply: This is corrected.*

Fig. 3 and 4 The colorscale is skewed (aspect ratio changed) also write "annual" in the caption

*Reply: we have done this in the revised version.*

Fig 5 and 6: The grey shading must be explained in the caption. Also I think the authors show some smoothed time series, I would not expect such a smooth annual time series from a model, please explain what it shown here. Is it a 1000 year running mean?

*Reply: We thank the reviewer for this important comment, we have now explained every detail in all the figures. Yes, the time series is a 1000-year running mean and we have now clarified this in the revised version.*

Fig7: Again the caption does not contain all explanations.

*Reply: We have now explained every detail in all the figures as suggested by the reviewer.*

L454 Still the sub section is only a few lines and the next covers the rest of section 4 so I suggest to avoid subsections in the discussion section.

*Reply: We have merged all the subsections in the discussion as suggested.*

L477-478: This is not enough. The authors need to discuss the seasonality in much more details. With their model simulations they can show results for the growing season fitting better to some of the proxy reconstructions.

*Reply: We thank the reviewer for this suggestion. We have compared our model's growing season (as expressed by Growing Degree Days 0) with that of the reconstructions of Mauri et al 2015 (supplementary information) and other proxy-based data and it fits better with some proxy reconstructions.*

L526 bracket missing.

Reply: *we have corrected this.*

L580: the statement "In any case, it is challenging

to determine which model would best depict the climate at the mid-Holocene" is awkward, please reformulate or remove.

*Reply: We have removed this sentence.*

L596 I recommend to include a discussion on the weaknesses/limitations of the method used. This is essential as at the moment the reader gets the impression that there are no weaknesses. So include this in the conclusion section maybe at the end and also in the discussion section.

*Reply: we have added this to the last paragraph of the discussion as suggested by the reviewer.*

L615 The sentence "reconstructions, we agree that that our 11.5K_Down simulates in some" makes no sense, please revise.

*Reply: We have made this correction.*

There are still errors in the reference list, e.g. "Davis et al., 2003 B.A.S. Davis, S. Brewer, A.C. Stevenson, J. Guiot.:" I will never understand why author still hard code references and not use endnote or other tools.

*Reply: We have made all the corrections in the reference list*